# NEAR: A Training-Free Pre-Estimator of Machine Learning Model Performance

**Raphael T. Husistein, Markus Reiher, and Marco Eckhoff**
Department of Chemistry and Applied Biosciences,
ETH Zurich,
Vladimir-Prelog-Weg 2, 8093 Zurich, Switzerland.
`{raphahus,mreiher,eckhoffm}@ethz.ch`

## Abstract

Artificial neural networks have been shown to be state-of-the-art machine learning models in a wide variety of applications, including natural language processing and image recognition. However, building a performant neural network is a laborious task and requires substantial computing power. Neural Architecture Search (NAS) addresses this issue by an automatic selection of the optimal network from a set of potential candidates. While many NAS methods still require training of (some) neural networks, zero-cost proxies promise to identify the optimal network without training. In this work, we propose the zero-cost proxy *Network Expressivity by Activation Rank* (NEAR). It is based on the effective rank of the pre- and post-activation matrix, i.e., the values of a neural network layer before and after applying its activation function. We demonstrate the cutting-edge correlation between this network score and the model accuracy on NAS-Bench-101 and NATS-Bench-SSS/TSS. In addition, we present a simple approach to estimate the optimal layer sizes in multi-layer perceptrons. Furthermore, we show that this score can be utilized to select hyperparameters such as the activation function and the neural network weight initialization scheme.

## 1 Introduction

Originally inspired by the structure and function of biological neural networks, artificial neural networks are now a key component in machine learning (Bishop, 2006; Russell & Norvig, 2021). They are applied in many tasks such as natural language processing, speech recognition, and even protein folding (Vaswani et al., 2017; Baevski et al., 2020; Jumper et al., 2021). Artificial neural networks are built by layers of interconnected neurons, which receive inputs, apply a, typically non-linear, activation function, and pass on the result (see Figure A.1). By tuning the weights of the individual inputs to a neuron, a neural network can learn functional relations from sample data to perform predictions without explicit program instructions. Deep learning is obtained if more than one layer is hidden between the first input and last output layer of the neural network. In theory, already a neural network with a single hidden layer and enough neurons is capable of approximating any continuous function up to arbitrary precision (Hornik et al., 1989; Cybenko, 1989; Hornik, 1991).

However, the choice of the model size, i.e., the number of hidden layers and the number of neurons per layer, has a huge impact on the performance, training time, and computational demand (Tan & Le, 2019). A too small network is unable to capture a complex relation in a dataset resulting in underfitting. Conversely, a too large model lacks efficiency and carries the risk of overfitting and poor generalization. Determining the optimal model size that strikes a balance between accuracy and efficiency remains an important challenge (Ren et al., 2021). In common practise, various models of different sizes are trained and their final performances are compared. This manual process is time-consuming and often leads to sub-optimal model architectures. The goal of Neural Architecture Search (NAS) is to eliminate this manual process and autonomously identify suitable architectures (Elsken et al., 2019). Commonly applied techniques for NAS are based on reinforcement learning, evolutionary algorithms, or gradient-based optimization (Zoph & Le, 2017; Real et al., 2017; Liu et al., 2019). However, these methods are still time- and resource-intensive. Zero-cost proxies have been proposed to improve the efficiency of NAS methods (Mellor et al., 2021; Abdelfattah

et al., 2021; Krishnakumar et al., 2022). They exploit properties of the untrained network that are correlated with the final accuracy, bypassing the costly training process. In this way, they promise speed-ups of several orders of magnitude compared to traditional NAS algorithms (Abdelfattah et al., 2021).

The performance of previously reported zero-cost proxies is highly dependent on the underlying search space (Ning et al., 2021). Furthermore, many of them are unable to demonstrate consistently a higher correlation with the final accuracy than the trivial baseline given by the number of parameters (#Params) (Ning et al., 2021). In addition, these proxies require a search space of known architectures. While the latter is not a severe limitation in applications such as computer vision, in many other applications such search spaces do not exist. An example is a so-called machine learning potential (Behler & Parrinello, 2007; Behler, 2021) applied in chemistry and materials science, which employs neural networks to predict the energy of a chemical system as a function of the molecular structure. In this work, we propose the zero-cost proxy *Network Expressivity by Activation Rank* (NEAR) that can be used in conjunction with a search space, but also provides the possibility to estimate the optimal layer size for multi-layer perceptrons without a search space.

Besides the network architecture, the activation function and weight initialization scheme have a significant impact on the training dynamics and the final performance of a neural network (LeCun et al., 1998; Glorot & Bengio, 2010). A variety of techniques have been developed to select such hyperparameters, including methods based on Bayesian optimization and evolutionary algorithms (Snoek et al., 2012; Lorenzo et al., 2017). However, these techniques still require expensive training of multiple networks, and the zero-cost proxies reported so far have not been applied to the selection of the activation function and the weight initialization scheme. A zero-cost proxy that can be applied not only to find optimal architectures, but also to choose the activation function and the weight initialization scheme, would allow one to further reduce the computational burden of constructing a performant artificial neural network. Here, we therefore evaluate the suitability of our zero-cost proxy NEAR for this purpose.

In summary, we make the following three key contributions: First, we propose the zero-cost proxy NEAR and test its performance across several search spaces and datasets. Second, we introduce a simple method to estimate the optimal layer size of multi-layer perceptrons and show its effectiveness for machine learning potentials. Third, we empirically demonstrate that our zero-cost proxy NEAR can support the selection of an activation function and a suitable weight initialization scheme.

## 2 RELATED WORK

Zero-cost NAS proxies are designed to identify optimal neural network architectures without requiring training. Some of the earliest proxies such as *Single-Shot Network Pruning Based on Connection Sensitivity* (SNIP), *Gradient Signal Preservation* (GraSP), Fisher, and *Synaptic Flow* (SynFlow) have been adapted from pruning techniques and are based on gradients of the neural network (Abdelfattah et al., 2021). While SNIP assigns scores to individual parameters by approximating the change in loss when the parameter is removed (Lee et al., 2019), the scores of GraSP are based on an approximation of the change in the gradient norm (Wang et al., 2020). The Fisher proxy estimates the importance of an activation by calculating its contribution to the loss (Turner et al., 2020), and SynFlow assigns a score to each parameter indicating how much it contributes to the information flow in the network (Tanaka et al., 2020). A proxy that has not been adapted from the pruning literature is Grad_norm, which is defined as the Euclidean norm of the gradient (Abdelfattah et al., 2021). SNIP, GraSP, SynFlow, and Grad_norm are calculated for each individual parameter, whereas Fisher is calculated for channels within a convolution layer. To obtain a score for the entire network, the values assigned to the parameters or channels are summed. *Zero-Shot NAS via Inverse Coefficient of Variation on Gradients* (ZiCo) (Li et al., 2023) is another proxy that exploits gradient information, linking the mean and the variance of the gradient to the convergence rate and the capacity of the neural network. With the exception of SynFlow, all the aforementioned proxies require the presence of target output, so-called labels. Therefore, they can only be applied to a limited extent in unsupervised learning for categorization tasks.

Some zero-cost proxies exist that employ properties of the untrained network that do not require backpropagation and hence work without labels. These include *Neural Architecture Search Without Training* (NASWOT) (Mellor et al., 2021), Zen-Score (Lin et al., 2021), *Sample-Wise Activation*

*Patterns* (swap) (Peng et al., 2024), *Regularized Sample-Wise Activation Patterns* (reg_swap) (Peng et al., 2024), and *Minimum Eigenvalue of Correlation* (MeCo$_{opt}$) (Jiang et al., 2023). To a certain extent, these proxies can be related to theoretical considerations regarding, for example, the functions that can be represented by a particular neural network or the convergence rate during training. A proxy that combines information from gradients with the output of the untrained network and works without labels is *Training-Free Neural Architecture Search* (TE-NAS) (Chen et al., 2021).

All these proxies assess the performance of a network based on the output of individual untrained layers. However, most of them, namely NASWOT, Zen-Score, swap, reg_swap, and TE-NAS, can only be applied if the Rectified Linear Unit (ReLU) (Nair & Hinton, 2010) activation function is employed. Our zero-cost proxy NEAR utilizes the output of individual layers before and after the application of the activation function for sample inputs without constraints on the type of activation function. Therefore, its applicability is more general.

## 3 METHODS

### 3.1 MULTI-LAYER PERCEPTRON

An artificial neural network with the architecture described in the introduction (see Figure A.1) is also called multi-layer perceptron. A multi-layer perceptron with $L$ layers can be expressed as

$$\mathcal{F}(\mathbf{x}) = (\sigma \circ \mathbf{W}_L) \circ (\sigma \circ \mathbf{W}_{L-1}) \circ \cdots \circ (\sigma \circ \mathbf{W}_2) \circ (\sigma \circ \mathbf{W}_1)(\mathbf{x}) := \mathbf{y} \,, \qquad (1)$$

where $\mathbf{x}$ and $\mathbf{y}$ are the input and output features, $\sigma$ is an activation function, and $\mathbf{W}_l \in \mathbb{R}^{N_l \times N_{l-1}}$ are the weights of layer $l$. The weights can also include so-called bias weights, which are added to the output of the neuron without being multiplied by any input value. Hence, they effectively shift the activation function input of the neuron.

We define the pre-activation of layer $l$ as

$$\mathbf{z}_l = (\mathbf{W}_l) \circ (\sigma \circ \mathbf{W}_{l-1}) \circ \cdots \circ (\sigma \circ \mathbf{W}_2) \circ (\sigma \circ \mathbf{W}_1)(\mathbf{x}) \qquad (2)$$

and the post-activation as

$$\mathbf{h}_l = \sigma(\mathbf{z}_l) \,. \qquad (3)$$

### 3.2 NETWORK EXPRESSIVITY BY ACTIVATION RANK (NEAR)

The construction of our zero-cost proxy NEAR is inspired by theoretical considerations on machine learning that attempt to assess the expressivity of artificial neural networks (Pascanu et al., 2013; Montufar et al., 2014). Therefore, we restate here the most important concepts from these works. For neural networks applying the ReLU activation function, each activation function can be conceptualized as a hyperplane that separates the input space into an active and an inactive region. Both regions are referred to as linear regions. The number of such linear regions can be viewed as a measure of the expressivity of the network (Pascanu et al., 2013; Montufar et al., 2014). Following previous studies (Raghu et al., 2017; Montúfar, 2017; Serra et al., 2018), we define the activation pattern for an input $\mathbf{x}$ and a network $\mathcal{F}$ with the ReLU activation function by the set of vectors $\mathcal{A}(\mathbf{x}; \mathcal{F}) = \{\mathbf{a}_1, \ldots, \mathbf{a}_L\}$ where $a_{li} = 1$ if the output of the $i$th neuron in layer $l$ is positive and $a_{li} = 0$ otherwise. Given that different linear regions correspond to distinct activation patterns, the number of linear regions is equivalent to the number of activation patterns. Consequently, it is a natural idea to analyze the activation patterns to assess the expressivity of a network. This idea is exploited in several zero-cost proxies (Mellor et al., 2021; Chen et al., 2021; Lin et al., 2021; Peng et al., 2024). However, constructing a proxy based on the aforementioned definition of the activation pattern will restrict its application to networks applying the ReLU activation function, a drawback that all those proxies have in common.

To build a more general zero-cost proxy, we relax the definition of the activation pattern to enable other activation functions: $\mathcal{A}(\mathbf{x}; \mathcal{F}) = \{\mathbf{h}_1, \ldots, \mathbf{h}_L\}$, with $\mathbf{h}_l$ defined in Eq. (3). While there is no longer a direct relationship to the number of linear regions, we argue that inputs with a very similar activation pattern are more challenging for the network to distinguish, and we leverage this argument to develop our zero-cost proxy. We define the pre-activation matrix for layer $l$ as $\mathbf{Z}_l \in \mathbb{R}^{n_l \times n_l}$, where $n_l$ is the number of neurons in layer $l$. Each of the $n_l$ rows of $\mathbf{Z}_l$ contains $\mathbf{z}_l$ for some

different sample input $\mathbf{x}_i$. In a similar manner, the post-activation matrix for layer $l$ is defined as $\mathbf{H}_l \in \mathbb{R}^{n_l \times n_l}$, where each row contains $\mathbf{h}_l$ for a different sample input $\mathbf{x}_i$. Intuitively, we would expect a well-performing network when the rows in the pre-/post-activation matrix are very different to each other, while a matrix with all similar rows would indicate a poorly performing network. As an example, we can consider a classification problem in which rows are highly similar yet belong to inputs of different classes. In this case even a minor perturbation can result in a change in the predicted class. Additionally, we would expect that a matrix, in which certain rows have entries that are much larger than the entries in other rows, will result in poorer performance than a matrix with entries of similar size. The reason for this behavior is that the differences in size can cause the predictions to become highly dependent on individual weights and hence on individual input values. These two relationships are captured by the effective rank, defined as follows:

**Definition 3.1** Effective Rank (Roy & Vetterli, 2007). *The effective rank of a matrix $\mathbf{A} \in \mathbb{C}^{M \times N}$, with $Q = \min\{M, N\}$, is given by*

$$\mathrm{erank}(\mathbf{A}) = \exp[H(p_1, \ldots, p_Q)] , \tag{4}$$

*where $H(p_1, \ldots, p_Q)$ is the Shannon entropy (Shannon, 1948),*

$$H(p_1, \ldots, p_Q) = -\sum_{k=1}^{Q} p_k \log(p_k) , \text{ with } p_k = \frac{\tilde{\sigma}_k}{\sum_{i=1}^{Q} \tilde{\sigma}_i} , \tag{5}$$

*where $p_k$ are normalized singular values and $\tilde{\sigma}_i$ are singular values.*

To interpret the effective rank of a matrix, we can think of the matrix as a linear transformation. The rank of the matrix indicates the number of dimensions of its range. By contrast, the effective rank contains information about the geometrical shaping of the transformation. A linear transformation that causes a strong stretching along one dimension, while all other dimensions remain unchanged, has a lower effective rank than a transformation that causes an equally strong stretching along all dimensions. The rank of the matrix of both transformations is, however, identical. In addition, a linear transformation that stretches each dimension equally is represented by a matrix with orthogonal rows, and rows that are orthogonal to each other could be considered maximally different. In other words, the effective rank measures what we have intuitively explained above as being indicators of good performance. Consequently, we propose the zero-cost proxy NEAR as the following:

**Definition 3.2** Network Expressivity by Activation Rank (NEAR). *The NEAR score of a neural network is given by*

$$s = \sum_{l=1}^{L} \mathrm{erank}(\mathbf{Z}_l) + \mathrm{erank}(\mathbf{H}_l) ,$$

*where $\mathbf{Z}_l$ and $\mathbf{H}_l$ denote the pre- and post-activation matrices, respectively.*

A higher NEAR score $s$ indicates a better performing network. The score depends on the input samples, which will usually be randomly selected. To prevent large, random fluctuations depending on the selected samples, we recommend to calculate the score multiple times and employ the average. On the NAS benchmarks, we calculated the average of 32 repetitions, while for the machine learning potential benchmarks, we employed 400 repetitions due to the more diverse dataset.

## 3.3 NEAR FOR CONVOLUTIONAL NEURAL NETWORKS

Convolutional Neural Networks (CNNs) are a neural network architecture particularly popular in the field of computer vision (Li et al., 2021). Our description here focuses on networks applying 2D convolutions. However, the approach can be extended to 3D convolutions as well. For 2D convolutions, the input has the dimensions $W \times H \times C$, where $C$ is the number of channels. For example, $C$ equals 3 for color images, since these consist of red, green, and blue color channels with dimension $W \times H$ (number of pixels in width $W$ and height $H$). To extract features from the input, a convolution operation is performed (see Figure A.2) using one or more filters with dimension $k \times k \times C$, where $k$ is an adjustable hyperparameter. These filters are essentially the weights that are learned during the training process. The results of the convolutions are summed, so that each filter results in a single output matrix, also known as a feature map. The output is passed to the next layer, where new filters are applied, allowing the network to learn increasingly complex features.

We employ the feature maps in the construction of the activation matrix to calculate the NEAR score for a CNN. A layer with $C$ input channels results in an output of the form $\mathbf{Z}_l \in \mathbb{R}^{W' \times H' \times C'}$, with $W' = W - k + 1$ and $H' = H - k + 1$ (in case of no padding and stride of one). $C'$ denotes the number of filters in the layer. Since this output is a 3D tensor, the NEAR score cannot be calculated directly. Instead, the output is first transformed into a matrix. We consider the number of filters to be analogous to the number of neurons in a fully connected layer. Therefore, the maximum possible NEAR score should be larger if there are more filters in a layer. This behavior could be achieved by flattening each feature map into a vector and then packing the vectors into a matrix. However, for an image with three channels and $32 \times 32$ pixels as input, this approach quickly results in a large matrix. For example, when using a filter of dimension $3 \times 3 \times 8$, the matrix already is of dimensions $7200 \times 7200$, and the effort to compute the NEAR score is no longer negligible. Therefore, we approximate this matrix for the calculation of the NEAR score. First, we flatten the output to get $\mathbf{Z}_l \in \mathbb{R}^{W' \cdot H' \times C'}$ (see Figure A.3). Second, to reduce the size of the first dimension of this matrix, a random selection of $C'$ contiguous rows is made, where the first row contains elements taken from the first row of the feature maps. This approach yields essentially a submatrix of the large matrix containing all the feature maps as vectors. We employ this submatrix as a proxy to compute the NEAR score. For a comparison between the rank calculated from the full matrix and the submatrix, see Figures A.8 and A.9.

## 4 Results and Discussion

### 4.1 Correlation of Effective Rank and Final Model Accuracy

To examine the performance of our zero-cost proxy NEAR, we evaluated it on the three standard cell-based NAS benchmarks NAS-Bench-101 (Ying et al., 2019), NATS-Bench-SSS, and NATS-Bench-TSS (Dong et al., 2021). We note that "cell-based" refers to the construction of the individual networks, which were obtained by placing stacks of repeated cells in a common skeleton (for details see Ying et al. (2019) and Dong et al. (2021)). For comparison, we report the two commonly employed rank correlation measures Kendall's $\tau$ (Kendall, 1938) and Spearman's $\rho$ (Spearman, 1904) also for twelve other zero-cost proxies. For Grad_norm, SNIP, GraSP, Fisher, and SynFlow we relied on the implementations of Abdelfattah et al. (2021) and for ZiCo (Li et al., 2023), MeCo$_{opt}$ (Jiang et al., 2023), swap (Peng et al., 2024), and reg_swap (Peng et al., 2024) we employed the respective paper's implementation. All results were recalculated by us. In some cases we observed small differences in the correlation coefficients compared to the values of the original publications. To demonstrate that NEAR is not only applicable to image classification tasks, we provide supplementary results on TransNAS-Bench-101-Micro (Duan et al., 2021) and HW-GPT-Bench (Sukthanker et al., 2024) in Section A.2.

**NATS-Bench-TSS**

The NATS-Bench-TSS search space consists of $15\,625$ neural network architectures trained on the datasets CIFAR-10, CIFAR-100 (Krizhevsky & Hinton, 2009), and ImageNet16-120 (Chrabaszcz et al., 2017). The correlation between the different proxies and the final test accuracy is shown in Table 1. Some of the previously reported proxies, such as Grad_norm, SNIP, and GraSP, yield a lower correlation than the number of parameters. The latter can be seen as trivial baseline for the zero-cost proxies. The more recently developed proxies ZiCo, MeCo$_{opt}$, swap, reg_swap, and our proxy NEAR demonstrate superior performance relative to this baseline. In general, MeCo$_{opt}$ achieves the highest correlation across all datasets for NATS-Bench-TSS. Nevertheless, the NEAR and reg_swap results are very close to those of MeCo$_{opt}$. However, the comparison to the reg_swap score is not entirely fair because it assumes that architectures with a parameter number close to the chosen parameter $\mu$ perform best, which may not be universally true. MeCo$_{opt}$ and NEAR are therefore more generally applicable.

**NATS-Bench-SSS**

For the larger NATS-Bench-SSS search space with $32\,768$ architectures trained on CIFAR-10, CIFAR-100, and ImageNet16-120, neither MeCo$_{opt}$ nor reg_swap consistently show a higher correlation with the final test accuracy than the number of parameters (Table 2). Only SynFlow, NEAR, and ZiCo are able to surpass the performance of this baseline on all three datasets, with SynFlow

demonstrating the highest correlation on CIFAR-10 and ImageNet16-120, and NEAR yielding the highest correlation on CIFAR-100.

**NAS-Bench-101**

The NAS-Bench-101 search space consists of $423\,624$ architectures trained on the CIFAR-10 dataset. As shown in Table 3, NEAR achieves the highest correlation with the final accuracy for both metrics. It significantly outperforms MeCo$_{opt}$, SynFlow, and reg_swap, which show competitive performance on NATS-Bench-TSS and NATS-Bench-SSS. The second highest correlation for NAS-Bench-101 is shown by the Zen-Score, closely followed by ZiCo. However, Zen-Score yields worse correlation than the number of parameters on NATS-Bench-TSS across all datasets.

Table 1: Kendall's $\tau$ and Spearman's $\rho$ correlation coefficients for different zero-cost proxies on the NATS-Bench-TSS benchmark. The highest correlation is highlighted in bold face.

| NATS-Bench-TSS | | | | | | |
|---|---|---|---|---|---|---|
| Dataset | CIFAR-10 | | CIFAR-100 | | IN16-120 | |
| Correlation / Proxy | $\tau$ | $\rho$ | $\tau$ | $\rho$ | $\tau$ | $\rho$ |
| Grad_norm (Abdelfattah et al., 2021) | 0.48 | 0.65 | 0.47 | 0.64 | 0.41 | 0.56 |
| SNIP (Abdelfattah et al., 2021) | 0.48 | 0.64 | 0.47 | 0.63 | 0.43 | 0.57 |
| GraSP (Abdelfattah et al., 2021) | 0.39 | 0.55 | 0.40 | 0.57 | 0.40 | 0.55 |
| Fisher (Abdelfattah et al., 2021) | 0.41 | 0.55 | 0.41 | 0.56 | 0.35 | 0.47 |
| SynFlow (Abdelfattah et al., 2021) | 0.58 | 0.77 | 0.57 | 0.76 | 0.49 | 0.67 |
| Zen-Score (Lin et al., 2021) | 0.32 | 0.43 | 0.31 | 0.42 | 0.32 | 0.44 |
| FLOPs | 0.58 | 0.75 | 0.55 | 0.73 | 0.51 | 0.68 |
| #Params | 0.58 | 0.75 | 0.55 | 0.73 | 0.49 | 0.66 |
| ZiCo (Li et al., 2023) | 0.58 | 0.78 | 0.60 | 0.80 | 0.57 | 0.76 |
| MeCo$_{opt}$ (Jiang et al., 2023) | **0.72** | **0.90** | **0.73** | **0.90** | **0.67** | **0.84** |
| swap (Peng et al., 2024) | 0.64 | 0.82 | 0.65 | 0.82 | 0.58 | 0.74 |
| reg_swap (Peng et al., 2024) | 0.71 | 0.88 | 0.69 | 0.87 | 0.62 | 0.79 |
| **NEAR** | 0.70 | 0.88 | 0.69 | 0.87 | 0.66 | **0.84** |

Table 2: Kendall's $\tau$ and Spearman's $\rho$ correlation coefficients for different zero-cost proxies on the NATS-Bench-SSS benchmark. The highest correlation is highlighted in bold face.

| NATS-Bench-SSS | | | | | | |
|---|---|---|---|---|---|---|
| Dataset | CIFAR-10 | | CIFAR-100 | | IN16-120 | |
| Correlation / Proxy | $\tau$ | $\rho$ | $\tau$ | $\rho$ | $\tau$ | $\rho$ |
| Grad_norm (Abdelfattah et al., 2021) | 0.53 | 0.72 | 0.38 | 0.54 | 0.52 | 0.70 |
| SNIP (Abdelfattah et al., 2021) | 0.63 | 0.82 | 0.43 | 0.61 | 0.61 | 0.79 |
| GraSP (Abdelfattah et al., 2021) | 0.30 | 0.43 | 0.10 | 0.15 | 0.33 | 0.48 |
| Fisher (Abdelfattah et al., 2021) | 0.47 | 0.65 | 0.30 | 0.43 | 0.43 | 0.60 |
| SynFlow (Abdelfattah et al., 2021) | **0.79** | **0.94** | 0.59 | 0.78 | **0.81** | **0.95** |
| Zen-Score (Lin et al., 2021) | 0.75 | 0.92 | 0.50 | 0.69 | 0.72 | 0.89 |
| FLOPs | 0.44 | 0.61 | 0.19 | 0.28 | 0.41 | 0.57 |
| #Params | 0.69 | 0.87 | 0.53 | 0.72 | 0.68 | 0.86 |
| ZiCo (Li et al., 2023) | 0.72 | 0.89 | 0.54 | 0.74 | 0.73 | 0.90 |
| MeCo$_{opt}$ (Jiang et al., 2023) | 0.74 | 0.91 | **0.62** | 0.81 | 0.61 | 0.80 |
| swap (Peng et al., 2024) | 0.48 | 0.67 | 0.23 | 0.34 | 0.43 | 0.60 |
| reg_swap (Peng et al., 2024) | 0.62 | 0.81 | 0.38 | 0.54 | 0.55 | 0.73 |
| **NEAR** | 0.74 | 0.91 | **0.62** | **0.82** | 0.76 | 0.92 |

To conclude, we provide a performance ranking of the zero-cost proxies across all datasets and search spaces. For each combination of search space and dataset, we order the zero-cost proxies according to their Spearman's $\rho$ value. We identify the position in this order as rank, whereby the proxy of highest correlation is assigned a rank of one. Subsequently, the average over all datasets for each NAS search space and for all search spaces is calculated. For example, NEAR achieves ranks three, one, and two on NATS-Bench-SSS for CIFAR-10, CIFAR-100, and ImageNet16-120, respectively. These rankings result in an average rank of two on this search space. Of the proxies tested, reg_swap, SynFlow, ZiCo, MeCo$_{opt}$, and NEAR are on average better than the baseline given by the number of parameters (Table 4). However, only NEAR and ZiCo yield consistently higher correlation than the number of parameters for each combination of search space and dataset, whereby NEAR outperforms ZiCo in every case. For each search space, NEAR is either the best or second best zero-cost proxy highlighting its broad applicability. Overall, NEAR achieves the best rank, followed by MeCo$_{opt}$ and ZiCo.

Table 3: Kendall's $\tau$ and Spearman's $\rho$ correlation coefficients for different zero-cost proxies on the NAS-Bench-101 benchmark. The highest correlation is highlighted in bold face.

| NAS-Bench-101 | | |
|---|---|---|
| Correlation ⟍ Proxy | $\tau$ | $\rho$ |
| Grad_norm (Abdelfattah et al., 2021) | $-0.17$ | $-0.25$ |
| SNIP (Abdelfattah et al., 2021) | $-0.12$ | $-0.17$ |
| GraSP (Abdelfattah et al., 2021) | $0.17$ | $0.25$ |
| Fisher (Abdelfattah et al., 2021) | $-0.19$ | $-0.28$ |
| SynFlow (Abdelfattah et al., 2021) | $0.25$ | $0.37$ |
| Zen-Score (Lin et al., 2021) | $0.47$ | $0.64$ |
| FLOPs | $0.30$ | $0.43$ |
| #Params | $0.30$ | $0.43$ |
| ZiCo (Li et al., 2023) | $0.45$ | $0.63$ |
| MeCo$_{opt}$ (Jiang et al., 2023) | $0.35$ | $0.49$ |
| swap (Peng et al., 2024) | $0.31$ | $0.43$ |
| reg_swap (Peng et al., 2024) | $0.30$ | $0.43$ |
| **NEAR** | **0.52** | **0.70** |

Table 4: The rank of different zero-cost proxies averaged over all datasets of NATS-Bench-TSS, NATS-Bench-SSS, and NAS-Bench-101 and across all search spaces. The best rank is highlighted in bold face.

| Average rank | | | | |
|---|---|---|---|---|
| Search space ⟍ Proxy | TSS | SSS | 101 | All |
| GraSP (Abdelfattah et al., 2021) | 9.33 | 12.00 | 7.00 | 10.14 |
| Fisher (Abdelfattah et al., 2021) | 10.00 | 9.67 | 10.00 | 9.86 |
| Grad_norm (Abdelfattah et al., 2021) | 7.67 | 8.33 | 9.00 | 8.14 |
| FLOPs | 5.67 | 11.00 | 5.00 | 7.86 |
| SNIP (Abdelfattah et al., 2021) | 8.00 | 6.67 | 8.00 | 7.43 |
| Zen-Score (Lin et al., 2021) | 11.00 | 4.00 | 2.00 | 6.71 |
| swap (Peng et al., 2024) | 3.33 | 9.67 | 5.00 | 6.29 |
| #Params | 6.33 | 5.00 | 5.00 | 5.57 |
| reg_swap (Peng et al., 2024) | 2.00 | 7.67 | 5.00 | 4.86 |
| SynFlow (Abdelfattah et al., 2021) | 5.33 | **1.67** | 6.00 | 3.86 |
| ZiCo (Li et al., 2023) | 3.67 | 3.67 | 3.00 | 3.57 |
| MeCo$_{opt}$ (Jiang et al., 2023) | **1.00** | 3.67 | 4.00 | 2.57 |
| **NEAR** | 1.67 | 2.00 | **1.00** | **1.71** |

## 4.2 ESTIMATION OF OPTIMAL LAYER SIZE

Inspired by previous work that suggested a power law scaling for the accuracy with respect to the model size for language models (Kaplan et al., 2020), we analyzed the NEAR score of a single layer of different sizes. We observed that the relative score, i.e., the score divided by the number of neurons, appears to be well represented by a simple power function of the form $f(s) = \alpha + \beta \cdot s^{\gamma}$, where $s$ is the NEAR score (see Figure A.4). This behavior is consistent for two different datasets. One is the balanced version of the extended MNIST (EMNIST) dataset (Cohen et al., 2017). It contains handwritten digits and letters, with an equal number of examples for each class. The other is a molecular dataset for a lifelong machine learning potential (lMLP). This dataset includes target energies and atomic forces for features given by molecular structures, which were obtained from various conformations in 42 different $S_N2$ reactions (8 600 structures) (Eckhoff & Reiher, 2023).

We argue that a decreasing relative NEAR score indicates that the network benefits less from additional neurons and, at a certain threshold, its performance is no longer limited by its size. Our experiments suggest a threshold based on the slope of the power function to work reasonably well. We therefore propose the following approach to estimate the size of individual layers: First, calculate the NEAR score for some different layer sizes. Second, fit a power function to the calculated relative scores. Third, calculate at which layer size the slope of the fit falls below a certain threshold and apply this layer size for the neural network. Since the NEAR score of each layer depends only on the previous layers, this approach can also be applied to networks with multiple layers. Starting with the first layer, the size of all layers can be determined iteratively.

**Machine Learning Potential**

We applied our approach to estimate the layer sizes on a three-hidden-layer machine learning potential, employing the dataset B of Eckhoff & Reiher (2023). Our method predicted a size of $52$, $58$, and $60$ for the first, second, and third hidden layers, respectively, with a threshold of $0.5\%$ of the slope at a layer size of $1$. These sizes are significantly smaller than applied in Eckhoff & Reiher (2023) (which are $102$, $61$, and $44$). While this change reduces the number of parameters from $22\,990$ to $13\,909$, the average test loss only increases from $0.012 \pm 0.003$ to $0.015 \pm 0.004$. This increase is reflected in a change of the root mean squared error for the energies from $(4.4 \pm 1.2)\,\mathrm{meV\,atom}^{-1}$ to $(4.6 \pm 1.2)\,\mathrm{meV\,atom}^{-1}$ and for the forces from $(100 \pm 8)\,\mathrm{meV\,\mathring{A}}^{-1}$ to $(109 \pm 8)\,\mathrm{meV\,\mathring{A}}^{-1}$. These numbers represent the mean and standard deviation over 20 individual machine learning potentials, for which the training was started from different randomly initialized neural networks. We note that the test loss depends not only on the capacity of the neural network, but also on the training process. However, NEAR only aims to capture the capacity of a network and does not take into account the training dynamics. Therefore, NEAR alone cannot be expected to determine the perfect size but rather to give a good estimate. Keeping the size of the first and second hidden layers at $52$ and $58$, respectively, and only varying the size of the third hidden layer showed that the predicted size achieves good performance. Even increasing the layer size to $100$ neurons did not improve the loss (Table 5). However, a slightly smaller layer of $40$ neurons would still be sufficient to achieve the same performance. We note that smaller networks require fewer computational resources and are therefore more efficient to evaluate. Repeating the experiment for a variable second hidden layer showed again that the predicted size has favorable characteristics in terms of accuracy and number of parameters. Reducing the size of the second layer to $38$ neurons (reduction in the number of parameters by $\sim 16\%$), results in an increase in the loss of $20\%$. Conversely, increasing the neuron number to $78$ (increase in the parameter number by $\sim 16\%$), yields a loss decrease of only $\sim 7\%$ (Table 5).

**Balanced Extended MNIST**

In a second experiment, we estimated the layer sizes for a two-hidden layer neural network on the balanced EMNIST dataset. Again, we applied a threshold of $0.5\%$ of the slope at a layer size of $1$. The resulting sizes were $200$ for both, first and second hidden layer. A comparison of the test loss to a network with a reduced second layer size of $120$, which decreases the parameter number by $\sim 10\%$, shows an loss increase of $\sim 4\%$. The aforementioned loss values correspond to a classification accuracy of $86.85\%$ and $87.30\%$. Conversely, an increase in the neuron number to $280$ for the second layer, which increases the parameter number by $\sim 9\%$, results in a decrease in the loss of around $1\%$ and an increase in the accuracy to $87.34\%$. This observation suggests that a size of $200$ represents a reasonable compromise between efficiency and accuracy (Table 6). To check whether also the prediction for the first layer was reasonable, we fixed the size of the second layer to $200$ and train networks with various sizes for the first layer (Table 6). A comparison of the test loss for a network with a size of $160$ ($\sim 20\%$ decrease in the parameter number) yields an increase in the loss of $\sim 1.3\%$ and a decrease in the accuracy from $87.30\%$ to $87.14\%$. Even when the size of the first layer was increased to $260$ ($\sim 30\%$ increase in the parameter number), the loss can only be reduced by $\sim 0.7\%$ and the accuracy is increased to $87.32\%$. Again, the predicted size appears to offer a good trade-off between the number of parameters and the accuracy.

Table 5: The test loss of trained lMLPs when only the size of the second or third hidden layer is changed. Marked in bold is the architecture predicted by our approach. In this and all following tables, mean and standard deviation (rounded up) of 20 experiment repetitions are reported.

| Variation of second layer | Test loss | #Params | Variation of third layer | Test loss | #Params |
|---|---|---|---|---|---|
| $52 - 18 - 60$ | $0.022 \pm 0.004$ | $9\,389$ | $52 - 58 - 20$ | $0.017 \pm 0.003$ | $11\,509$ |
| $52 - 38 - 60$ | $0.018 \pm 0.004$ | $11\,649$ | $52 - 58 - 40$ | $0.015 \pm 0.003$ | $12\,709$ |
| $\mathbf{52 - 58 - 60}$ | $\mathbf{0.015 \pm 0.004}$ | $\mathbf{13,909}$ | $\mathbf{52 - 58 - 60}$ | $\mathbf{0.015 \pm 0.004}$ | $\mathbf{13,909}$ |
| $52 - 78 - 60$ | $0.014 \pm 0.004$ | $16\,169$ | $52 - 58 - 80$ | $0.014 \pm 0.003$ | $15\,109$ |
| $52 - 98 - 60$ | $0.014 \pm 0.003$ | $18\,429$ | $52 - 58 - 100$ | $0.015 \pm 0.003$ | $16\,309$ |

### 4.3 Estimation of Activation Function and Weight Initialization Performance

Numerous NAS benchmarks, including NATS-Bench-TSS and -SSS and NAS-Bench-101, apply the ReLU activation function for all networks. There is a lack of attention regarding zero-cost proxies

Table 6: The test loss on the balanced EMNIST dataset when only the size of the first or second hidden layer is changed. Marked in bold is the architecture predicted by our approach.

| Variation of first layer | Test loss | #Params | Variation of second layer | Test loss | #Params |
|---|---|---|---|---|---|
| $120 - 200$ | $0.393 \pm 0.012$ | $127\,847$ | $200 - 40$ | $0.433 \pm 0.017$ | $166\,967$ |
| $140 - 200$ | $0.380 \pm 0.006$ | $147\,547$ | $200 - 80$ | $0.397 \pm 0.011$ | $176\,887$ |
| $160 - 200$ | $0.373 \pm 0.009$ | $167\,247$ | $200 - 120$ | $0.383 \pm 0.009$ | $186\,807$ |
| $180 - 200$ | $0.368 \pm 0.007$ | $186\,947$ | $200 - 160$ | $0.376 \pm 0.009$ | $196\,727$ |
| $\mathbf{200 - 200}$ | $\mathbf{0.368 \pm 0.006}$ | $\mathbf{206,647}$ | $\mathbf{200 - 200}$ | $\mathbf{0.368 \pm 0.006}$ | $\mathbf{206,647}$ |
| $220 - 200$ | $0.363 \pm 0.007$ | $226\,347$ | $200 - 240$ | $0.364 \pm 0.006$ | $216\,567$ |
| $240 - 200$ | $0.359 \pm 0.006$ | $246\,047$ | $200 - 280$ | $0.364 \pm 0.008$ | $226\,487$ |
| $260 - 200$ | $0.366 \pm 0.007$ | $265\,747$ | $200 - 320$ | $0.363 \pm 0.009$ | $236\,407$ |
| $280 - 200$ | $0.363 \pm 0.008$ | $285\,447$ | $200 - 360$ | $0.359 \pm 0.007$ | $246\,327$ |

to detect favorable activation functions or weight initialization schemes. There are even proxies specifically designed for the ReLU activation function (Mellor et al., 2021; Chen et al., 2021; Peng et al., 2024). However, it is well known that choosing an appropriate activation function and weight initialization scheme is crucial for achieving good model performance (LeCun et al., 1998; Glorot & Bengio, 2010; Eckhoff & Reiher, 2023). To assess whether NEAR can be employed to select an activation function and a weight initialization scheme, we calculated the score before training and compared it with the final model performance.

**Machine Learning Potential**

Eckhoff & Reiher (2023) previously reported that the activation function $\mathrm{sTanh}(x) = 1.59223 \cdot \tanh(x)$ with a tailored weight initialization scheme improves the final accuracy of a lMLP. We repeated training for the same dataset and the same neural network architecture (number of neurons per layer: $133/102/61/44/1$) employing the CoRe optimizer (Eckhoff & Reiher, 2023; 2024a;b). Table 7 shows that the sTanh activation function with the tailored weight initialization scheme achieves the lowest test loss, while also exhibiting the largest NEAR score. Comparing the two weight initialization schemes sTanh and Kaiming uniform (He et al., 2015) demonstrates that the former improves performance and that the NEAR score can be applied as a reliable predictor for the more suitable weight initialization scheme. A comparison of the loss and NEAR score of different activation functions applying the same weight initialization scheme also shows that the score can guide the selection of an appropriate activation function. For example, the activation function sTanh with the customized initialization scheme exhibits the lowest loss and the highest NEAR score, while the activation function Tanhshrink shows the highest loss and lowest NEAR score for this weight initialization scheme. However, the discrepancies in loss for the three best activation functions sTanh, Sigmoid Linear Unit (SiLU) (Elfwing et al., 2018), and Tanh are minimal, whereas the differences in the NEAR score would suggest larger differences. The lowest NEAR score is obtained by Tanhshrink with Kaiming initialization, which is also consistent with the fact that this combination achieves the highest average loss. Hence, NEAR is capable of identifying more and less effective activation functions, while the ordering in close cases cannot be guaranteed. Since the loss function of the lMLP includes derivative constraints, the second derivative of the activation function appears in the gradient calculation. Consequently, only activation functions with a non-zero second derivative can be considered, resulting in the omission of the ReLU function in this experiment. This issue once again highlights the benefit of NEAR that it can be applied to any activation function.

**Balanced Extended MNIST**

To show that the NEAR score can guide the selection of the activation function and weight initialization scheme across various datasets and independent of the optimizer, we report the test loss and the NEAR score on the balanced EMNIST dataset. We employed a multi-layer perceptron with two hidden layers of size 200 and the Adam optimizer (Kingma & Ba, 2015). We applied early stopping with a patience of 10 epochs, whereby $10\%$ of the training set served as validation set. We report the test loss and the NEAR score for the initialization schemes Xavier uniform (Glorot & Bengio, 2010), Kaiming uniform, and uniform and the activation functions SiLU, ReLU, Tanh, and Tanhshrink in Table 8. The NEAR score indicates that the Xavier initialization is the most effective, followed by the Kaiming initialization and the uniform initialization. While the uniform initialization is undoubtedly the least favorable in terms of resulting test loss, the distinction between Xavier and Kaiming is not as clear-cut. For the activation functions under consideration, the average loss is found to be lower with the Kaiming initialization than the Xavier initialization, but the difference is marginal

and the standard deviations are higher than the difference. For the Kaiming and Xavier initializations, the activation functions SiLU, ReLU, and Tanh result in similar test losses, while the NEAR score gives the impression that there is a clear ordering for a given initialization. As the orderings of the test losses and NEAR scores do not agree for both initializations, we conclude that these small differences in the resulting loss cannot be captured by the NEAR score. We note here again that the NEAR score only accounts for the expressivity of the network and does not take into account the training process. Still, the NEAR score can reliably predict the worst performing activation function (Tanhshrink). For this activation function, the test loss is also significantly different from those of the others. Analogously, the NEAR score can correctly identify that SiLU, Tanhshrink, and ReLU lead to a lower loss than Tanh if the uniform initialization is applied. Hence, NEAR can predict the performance of an activation function and weight initialization scheme, though it may not resolve minor differences in the resulting loss. In addition, Section A.3 shows that NEAR can identify suitable activation functions and weight initialization schemes better than previous proxies.

Table 7: Comparison of the test loss (after training) and the NEAR score (before training) for various activation functions $\sigma$ and weight initialization schemes.

| $\sigma$, initialization | Test loss | NEAR score |
|---|---|---|
| sTanh, sTanh | $0.012 \pm 0.003$ | $70.8 \pm 0.2$ |
| SiLU, sTanh | $0.013 \pm 0.004$ | $63.1 \pm 0.3$ |
| Tanh, sTanh | $0.015 \pm 0.003$ | $67.2 \pm 0.3$ |
| Tanhshrink, sTanh | $0.027 \pm 0.006$ | $45.2 \pm 0.2$ |
| SiLU, Kaiming | $0.031 \pm 0.008$ | $57.1 \pm 2.5$ |
| sTanh, Kaiming | $0.041 \pm 0.019$ | $60.3 \pm 2.5$ |
| Tanh, Kaiming | $0.050 \pm 0.022$ | $58.5 \pm 2.3$ |
| Tanhshrink, Kaiming | $0.082 \pm 0.014$ | $38.7 \pm 1.8$ |

Table 8: Comparison of the test loss (after training) and the NEAR score (before training) for various activation functions $\sigma$ and weight initialization schemes on the balanced EMNIST dataset. For "SiLU, uniform" one outlier was excluded from the mean and standard deviation.

| $\sigma$, initialization | Test loss | NEAR score |
|---|---|---|
| SiLU, Kaiming | $0.367 \pm 0.008$ | $382.8 \pm 5.0$ |
| SiLU, Xavier | $0.368 \pm 0.008$ | $417.7 \pm 0.6$ |
| ReLU, Kaiming | $0.372 \pm 0.010$ | $403.7 \pm 7.4$ |
| ReLU, Xavier | $0.374 \pm 0.008$ | $421.1 \pm 0.8$ |
| Tanh, Kaiming | $0.392 \pm 0.010$ | $398.2 \pm 5.1$ |
| Tanh, Xavier | $0.396 \pm 0.011$ | $415.5 \pm 0.4$ |
| Tanhshrink, Kaiming | $0.447 \pm 0.022$ | $278.4 \pm 3.2$ |
| Tanhshrink, Xavier | $0.449 \pm 0.018$ | $401.4 \pm 1.2$ |
| SiLU, uniform | $0.525 \pm 0.040$ | $13.3 \pm 0.1$ |
| Tanhshrink, uniform | $0.602 \pm 0.030$ | $13.4 \pm 0.1$ |
| ReLU, uniform | $0.651 \pm 0.047$ | $13.3 \pm 0.1$ |
| Tanh, uniform | $3.858 \pm 0.030$ | $9.0 \pm 0.1$ |

## 5 CONCLUSION

In this work, we have introduced NEAR which is a zero-cost proxy to pre-estimate the performance of a machine learning model without training. For this purpose, the NEAR score estimates the expressivity of neural networks by calculating the effective rank of the pre- and post-activation matrix. We have demonstrated its strong correlation with the final model accuracy on NATS-Bench-SSS/TSS and NAS-Bench-101. In a ranking of 13 zero-cost proxies by their Spearman's correlation coefficient, NEAR achieves an average rank of 1.7 across all datasets and search spaces, while the second-best proxy $MeCo_{opt}$ has only an average rank of 2.6. In contrast to other proxies, NEAR only requires sample input data, but no output labels and it is not bound to a specific activation function. In fact, we have shown that it can even be applied to choose a well performing activation function and weight initialization scheme. Moreover, NEAR is not limited to a given search space and can be applied to estimate the optimal layer size in multi-layer perceptrons. This task represents a significant challenge in the construction of neural networks, and hence, NEAR can help substantially reduce invested human time and computational demand.

## ACKNOWLEDGMENT

This work was created as part of NCCR Catalysis (grant number 180544), a National Centre of Competence in Research funded by the Swiss National Science Foundation. M.E. was supported by an ETH Zurich Postdoctoral Fellowship.

## REPRODUCIBILITY STATEMENT

In order to make our calculations reproducible, we provide the code including all hyperparameter settings as well as the raw data on Zenodo (Husistein et al., 2024). The NEAR software is available on GitHub (`https://github.com/ReiherGroup/NEAR`).

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

## A APPENDIX

### A.1 SUPPORTING FIGURES

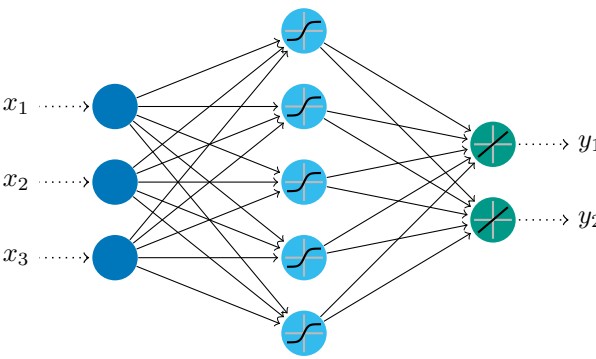

Figure A.1: Illustration of an artificial neural network with three inputs $\{x_i\}$ and two outputs $\{y_i\}$. The single hidden layer consists of five neurons. The activation functions are shown within the neurons. The solid lines represent the weights, while the dashed lines indicate input and output without weights.

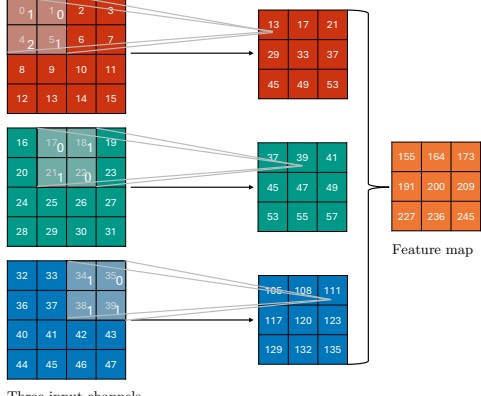

Figure A.2: A convolution is performed on the input of dimension $4 \times 4 \times 3$ with a filter of dimension $2 \times 2 \times 3$ resulting in a feature map of dimension $3 \times 3$.

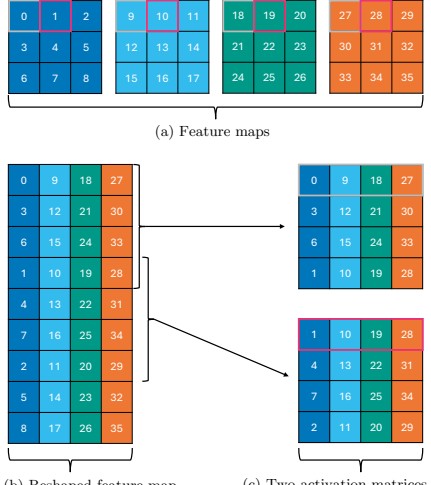

Figure A.3: Process of reshaping convolutional neural network feature maps. (a) The process begins with four $3 \times 3$ feature maps. (b) These feature maps are subsequently reshaped to a $9 \times 4$ matrix. (c) An activation matrix is given by four contiguous rows, whereby the first row contains elements extracted from the top row of the feature maps. From all possible activation matrices one is randomly selected.

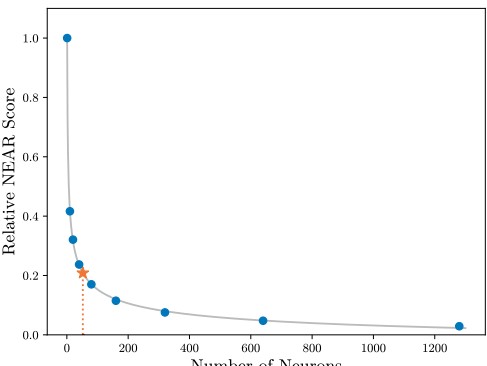

Figure A.4: A power function fitted to the NEAR score divided by the total number of neurons in the layer. The star marks the first time where the slope is smaller or equal to $0.5\%$ of the slope at $x = 1$. The plot has been generated for the experiments on the lMLP.

## A.2 ADDITIONAL NAS-BENCHMARKS

**TransNAS-Bench-101-Micro**

To demonstrate that NEAR is not only applicable to image classification tasks, we show here results on seven different tasks from the TransNAS-Bench-101-Micro benchmark (Duan et al., 2021). The tasks include classification, regression, pixel-level prediction, and self-supervised tasks. NEAR outperforms the baseline given by the number of parameters and the number of FLOPs on each task (Table 9).

Table 9: Kendall's $\tau$ and Spearman's $\rho$ correlation coefficients for NEAR on different tasks of TransNAS-Bench-101-Micro.

| Task
Proxy | Cls. Object | | Cls. Scene | | Autoencoding | | Surf. Normal | | Sem. Segment. | | Room Layout | | Jigsaw | |
|---|---|---|---|---|---|---|---|---|---|---|---|---|---|---|
| | $\tau$ | $\rho$ | $\tau$ | $\rho$ | $\tau$ | $\rho$ | $\tau$ | $\rho$ | $\tau$ | $\rho$ | $\tau$ | $\rho$ | $\tau$ | $\rho$ |
| #Params | 0.36 | 0.53 | 0.44 | 0.62 | −0.03 | −0.04 | 0.46 | 0.65 | 0.45 | 0.63 | 0.24 | 0.38 | 0.30 | 0.45 |
| FLOPs | 0.37 | 0.55 | 0.45 | 0.64 | −0.03 | −0.04 | 0.47 | 0.66 | 0.46 | 0.65 | 0.24 | 0.39 | 0.31 | 0.46 |
| **NEAR** | **0.54** | **0.74** | **0.60** | **0.80** | **0.14** | **0.22** | **0.66** | **0.84** | **0.57** | **0.76** | **0.37** | **0.55** | **0.45** | **0.64** |

**HW-GPT-Bench**

One of the strengths of NEAR is its ability to work with networks which do not employ the ReLU activation function. To explore this capability, we applied NEAR to language models with the Gaussian Error Linear Unit (GELU) (Hendrycks & Gimpel, 2016) activation function using the HW-GPT benchmark (Sukthanker et al., 2024). Specifically, we randomly sampled $20\,000$ architectures from the small variant of the search space and computed the correlation between the NEAR score and perplexity. While NEAR did not outperform the strong baseline given by the correlation between model parameters and perplexity, it still showed a higher correlation than the $\text{MeCo}_{\text{opt}}$ proxy (Table 10).

Table 10: Kendall's $\tau$ and Spearman's $\rho$ correlation between zero-cost proxy score and perplexity for $20\,000$ randomly sampled architectures of the small HW-GPT-Bench search space. As lower perplexity values indicate better models, the correlations are expected to be negative.

| Correlation
Proxy | $\tau$ | $\rho$ |
|---|---|---|
| #Params | $-0.92$ | $-0.99$ |
| FLOPs | $-0.92$ | $-0.99$ |
| $\text{MeCo}_{\text{opt}}$ (Jiang et al., 2023) | $-0.68$ | $-0.88$ |
| **NEAR** | $\mathbf{-0.90}$ | $\mathbf{-0.99}$ |

## A.3 Proxies to Select Weight Initialization and Activation Function

To show that NEAR outperforms previous proxies in identifying suitable weight initialization schemes and activation functions, we calculated the proxy scores for the same combinations listed in Table 8. Subsequently, we calculated Spearman's $\rho$ and Kendall's $\tau$ correlation coefficients for the average accuracy as a function of the average proxy score. The results indicate that NEAR exhibits a significantly higher correlation compared to the other proxies evaluated (see Table 11). Since the swap and reg_swap scores are only defined for the ReLU activation function, they have been omitted from this analysis.

Table 11: Kendall's $\tau$ and Spearman's $\rho$ correlation coefficients between the average model accuracy and the average proxy scores evaluated on the combinations of activation functions and weight initialization methods listed in Table 8. Averages are obtained from 20 repetitions. A higher correlation means that the proxy is more effective at identifying good activation functions and weight initializations.

| Correlation
Proxy | $\tau$ | $\rho$ |
|---|---|---|
| Grad_norm (Abdelfattah et al., 2021) | $-0.53$ | $-0.32$ |
| SNIP (Abdelfattah et al., 2021) | $-0.53$ | $-0.32$ |
| GraSP (Abdelfattah et al., 2021) | $0.53$ | $0.32$ |
| Fisher (Abdelfattah et al., 2021) | $-0.53$ | $-0.35$ |
| SynFlow (Abdelfattah et al., 2021) | $-0.15$ | $-0.08$ |
| Zen-Score (Lin et al., 2021) | $-0.35$ | $-0.24$ |
| ZiCo (Li et al., 2023) | $-0.07$ | $0.00$ |
| $\text{MeCo}_{\text{opt}}$ (Jiang et al., 2023) | $0.67$ | $0.45$ |
| **NEAR** | $\mathbf{0.84}$ | $\mathbf{0.70}$ |

## A.4 Stability of NEAR Score After Minimal Training

Similar to the analysis in (Mok et al., 2022), we computed the NEAR score after 0, 1, 3, 5, and 10 epochs of training on the NATS-Bench-SSS benchmark using the CIFAR-10 dataset. While many of the tested proxies show a decrease in correlation after some training, the correlation of NEAR only decreases slightly after the first training epoch and increases steadily for the following epochs (Figure A.5). Remarkably, NEAR is the only proxy that achieves a higher correlation after ten epochs of training than it had prior to training.

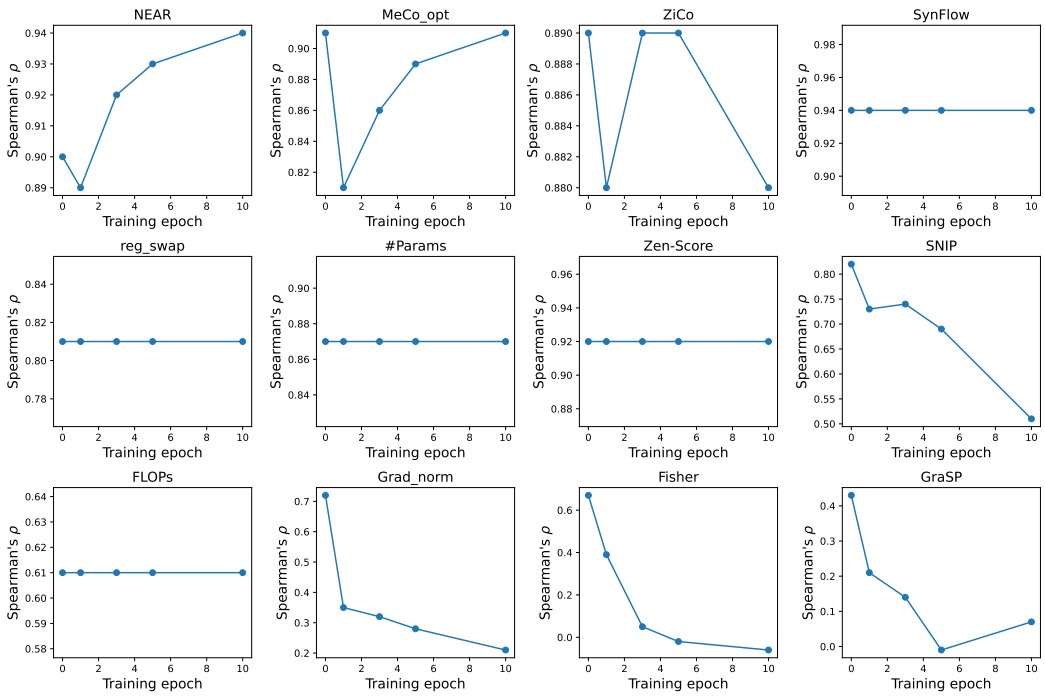

Figure A.5: Spearman's $\rho$ correlation on NATS-Bench-SSS using the CIFAR-10 dataset, evaluated after 0, 1, 3, 5, and 10 training epochs.

## A.5 INTERPRETATION OF CORRELATIONS

To provide further insight into the interpretation of the correlations between proxy scores and accuracy, we calculated the probability that, given two randomly selected networks $N_1$ and $N_2$, if the accuracy of $N_1$ is higher than that of $N_2$, then the proxy score of $N_1$ will also be higher than the proxy score of $N_2$. These probabilities are presented in Table 12.

| Search Space | NATS-Bench-TSS | | | NATS-Bench-SSS | | | NAS-Bench-101 |
|---|---|---|---|---|---|---|---|
| Correlation
Proxy | CIFAR-10 | CIFAR-100 | IN16-120 | CIFAR-10 | CIFAR-100 | IN16-120 | CIFAR-10 |
| swap (Peng et al., 2024) | 0.82 | 0.82 | 0.79 | 0.74 | 0.62 | 0.72 | 0.62 |
| reg_swap (Peng et al., 2024) | 0.85 | 0.85 | 0.81 | 0.81 | 0.69 | 0.77 | 0.65 |
| MeCo$_{opt}$ (Jiang et al., 2023) | **0.86** | **0.86** | **0.84** | 0.87 | **0.81** | 0.81 | 0.67 |
| ZiCo (Li et al., 2023) | 0.79 | 0.80 | 0.79 | 0.86 | 0.77 | 0.86 | 0.73 |
| Zen-Score (Lin et al., 2021) | 0.66 | 0.66 | 0.66 | 0.88 | 0.75 | 0.86 | 0.73 |
| SynFlow (Abdelfattah et al., 2021) | 0.79 | 0.78 | 0.75 | **0.90** | 0.79 | **0.91** | 0.63 |
| Fisher (Abdelfattah et al., 2021) | 0.70 | 0.71 | 0.68 | 0.74 | 0.65 | 0.72 | 0.41 |
| GraSP (Abdelfattah et al., 2021) | 0.69 | 0.70 | 0.70 | 0.65 | 0.55 | 0.67 | 0.59 |
| SNIP (Abdelfattah et al., 2021) | 0.74 | 0.73 | 0.71 | 0.82 | 0.72 | 0.80 | 0.44 |
| Grad_norm (Abdelfattah et al., 2021) | 0.74 | 0.74 | 0.71 | 0.77 | 0.69 | 0.76 | 0.42 |
| #Params | 0.73 | 0.72 | 0.71 | 0.85 | 0.77 | 0.84 | 0.65 |
| FLOPs | 0.73 | 0.72 | 0.71 | 0.72 | 0.60 | 0.71 | 0.65 |
| **NEAR** | 0.85 | 0.84 | 0.83 | 0.87 | **0.81** | 0.88 | **0.76** |

Table 12: The probability that for two randomly chosen networks the network with the higher accuracy also shows the higher zero-cost proxy score. This probability was estimated by randomly sampling one million pairs of networks. The highest probability is highlighted in bold face.

## A.6 WORST PREDICTIONS

An examination of NEAR's 100 most inaccurate predictions reveals that its main challenge is in evaluating networks with a large number of parameters. However, unlike other proxies, NEAR does not misclassify a network with low accuracy as one with high accuracy, thus reliably excluding networks with poor performance (see Figure A.6).

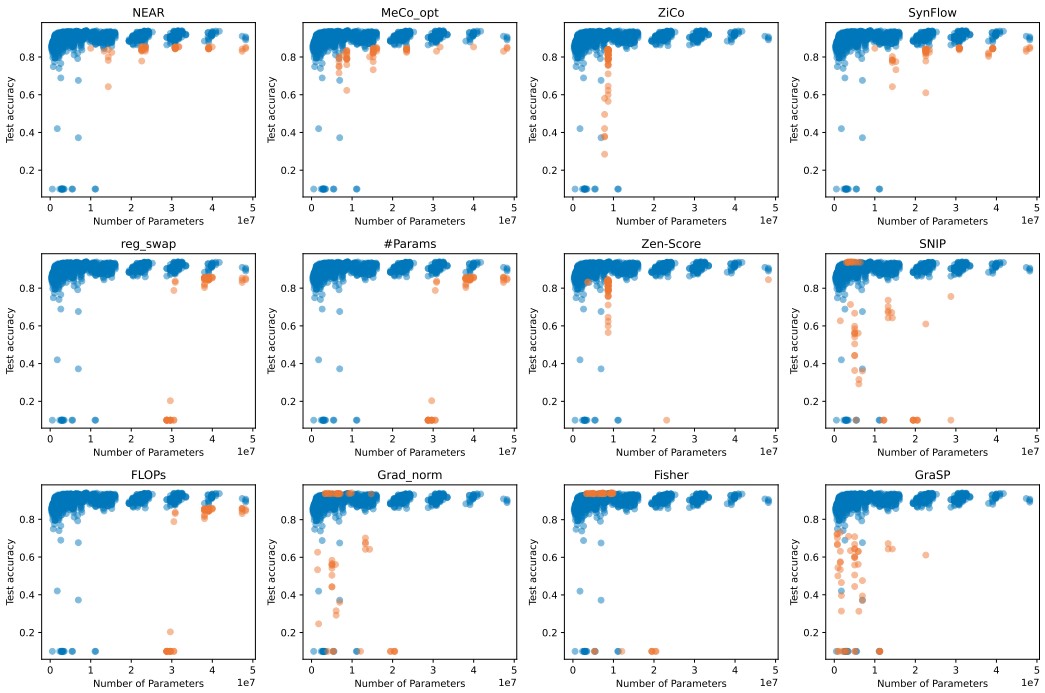

Figure A.6: Each point represents a neural network from the NAS-Bench-101 benchmark. The points highlighted in orange indicate the 100 neural networks where the proxy's predictions deviated most significantly from actual performance.

## A.7 CONTRIBUTIONS OF INDIVIDUAL LAYERS

To investigate how individual layers contribute to the overall correlation, we performed a layer-specific correlation analysis. This analysis involved generating over a billion random subsets, each comprising 6 layers. For each subset, we calculated the NEAR score. Then, for each layer, we summed the NEAR scores from all subsets containing that layer. Finally, we divided this sum by the total number of subsets in which each layer appeared. This analysis revealed no layer with a significantly lower or higher average correlation than the others. In other words, the results indicated no individual layer exerted a disproportionately strong or weak influence on the overall correlation. The average correlation per layer is shown in Figure A.7.

## A.8 SAMPLING OF FEATURE MAPS

In Section 3.3, we explained how NEAR can be calculated using only a subsample of the full matrix, which consists of all feature maps represented as vectors (see also Figure A.3). For NEAR to yield meaningful scores despite this approximation, the subsample matrix needs to remain sensitive to variations in kernel size and the number of channels. As shown in Figure A.8 and Figure A.9, the effective rank derived from both, the full matrix and the subsample matrix, exhibit similar trends, indicating that these parameters are accurately captured even within the approximation. In addition, Table 13 shows the impact of the sample size on the estimation of the rank of the full matrix.

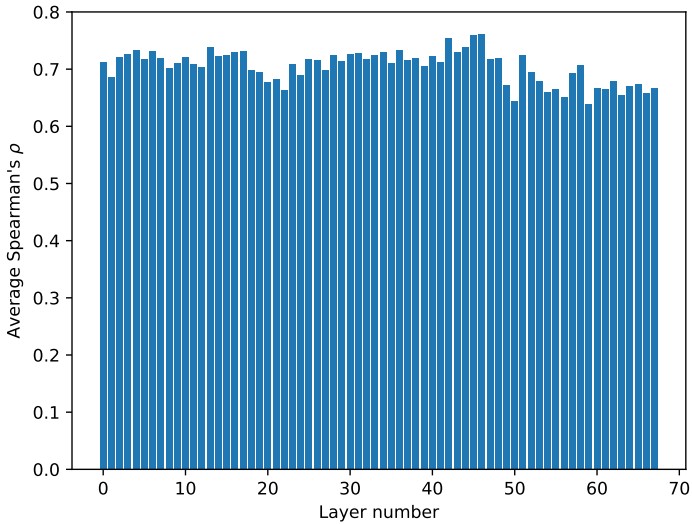

Figure A.7: The average correlation for each layer on NATS-Bench-SSS for CIFAR-10, determined by computing the correlations for random subsets of six layers and then averaging the results.

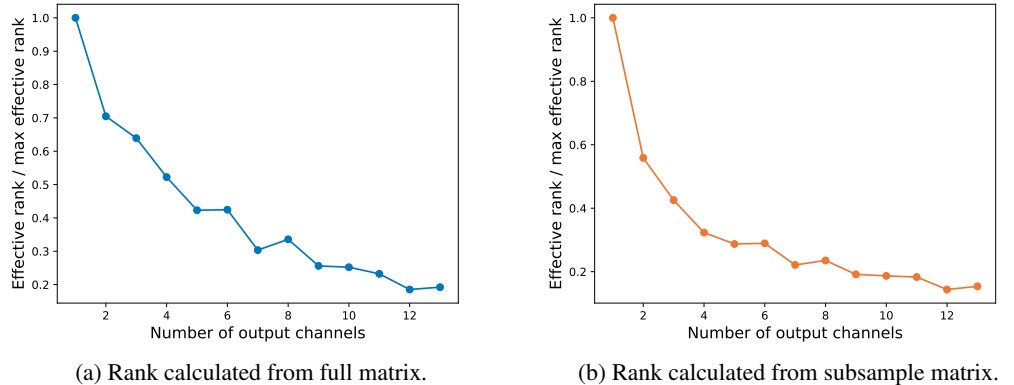

(a) Rank calculated from full matrix.      (b) Rank calculated from subsample matrix.

Figure A.8: Comparison of the rank calculated from the full matrix and the subsample matrix as a function of the number of channels. This evaluation is based on the CIFAR-10 dataset.

| # Samples \ Parameter | Kernel size | Channel number |
|---|---|---|
| 1 | 0.95 | 0.92 |
| 10 | 0.98 | 0.98 |
| 32 | 0.98 | 0.99 |

Table 13: The first column shows Spearman's $\rho$ correlation between the rank calculated from the full matrix and the rank calculated from the subsample matrix employing kernel sizes of 1, 3, 5, 7, and 9 averaged over $10\,000$ iterations. The second column shows Spearman's $\rho$ correlation between the rank calculated from the full matrix and the rank calculated from the subsample matrix employing between 1 and 13 output channels averaged over $10\,000$ iterations.

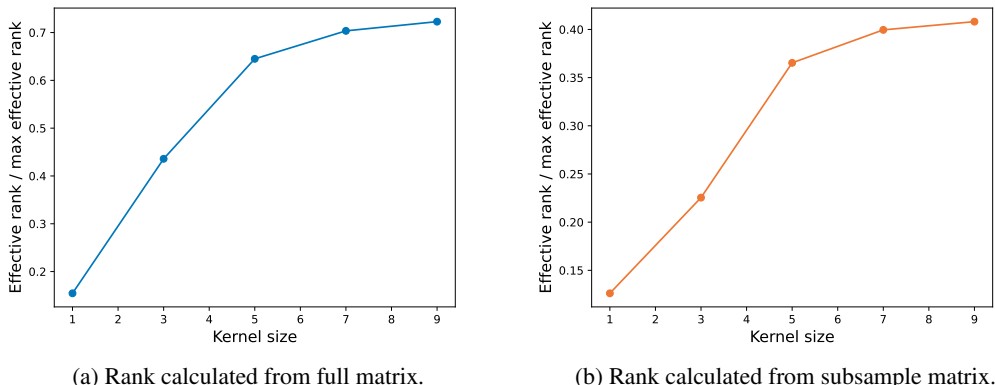

(a) Rank calculated from full matrix.

(b) Rank calculated from subsample matrix.

Figure A.9: Comparison of the rank calculated from the full matrix and the subsample matrix as a function of the kernel size. This evaluation is based on the CIFAR-10 dataset.

