# OpenReview forum: "NEAR: A Training-Free Pre-Estimator of Machine Learning Model Performance"
_ICLR.cc/2025/Conference — ICLR 2025 Poster_

### Official Review · Reviewer_o26N · 2024-11-03

**Soundness:** 3
**Presentation:** 3
**Contribution:** 3
**Rating:** 8
**Confidence:** 4

**Summary:**

NEAR is a zero-cost proxy that predicts machine learning model performance without training by estimating neural network expressivity through the effective rank of activation matrices. It demonstrates a strong correlation with final model accuracy on NATS-Bench and NAS-Bench-101, ranking first among 13 zero-cost proxies. NEAR requires only input data, not labels, and works with any activation function. It also helps select optimal activation functions, weight initializations, and layer sizes in multi-layer perceptrons.

**Strengths:**

The paper attempts to address a well-established and critical problem of correctly predicting the accuracies of different model architectures using zero-cost proxies.

Applies a good intuition of using ranks of pre and post-activation matrices as indicators of accuracy.

Novel analysis of weight initialization and activation functions.

Comprehensive comparison with existing state-of-the-art on relevant benchmarks.

**Weaknesses:**

Although the authors show good performance of their techniques across benchmarks, it is not clear if the metrics used to determine the model performance compare different architectures correctly. For example, given two model architectures A1 and A2 with test accuracy of A1 being greater than A2,  how often is the case where the rank of pre and post-matrices of A1 greater than A2?

How well does this generalize to larger models and more complex datasets?

For convolution operations, there are many hyperparameters like kernel size, stride, dilation, in-channels, out-channels, grouped convolutions, etc. How does the flattening of these matrices and calculating the score take into account all of the above factors? The intuition in this case is unclear.

**Questions:**

In the case of using ranks of pre and post-activation matrices as indicators of accuracy, a study that verifies this claim would be helpful. The same goes for the convolution part.

Specifically for the convolution part how accurate is the predicted rank using sampling? What is the minimum sample size required?

For the results, in cases where the estimations are off, why does that happen? A post-analysis would provide some insights into cases where NEAR is not optimal

It has been shown in the literature that randomly generated matrices in high dimensions contain almost linearly independent vectors, how does this impact the prediction if we only derive the rank hypothesis only from a few input samples?

---

> ### Author Response · Authors · 2024-11-26
>
> - *"Although the authors show good performance of their techniques across benchmarks, it is not clear if the metrics used to determine the model performance compare different architectures correctly. For example, given two model architectures A1 and A2 with test accuracy of A1 being greater than A2, how often is the case where the rank of pre and post-matrices of A1 greater than A2?"*
>
> To address your concern, we have now included an additional analysis to clarify the relationship between test accuracy and the ranking of pre- and post-activation metrics. Specifically, we have added Table 12 in the revised manuscript, which shows the probability that if model architecture A1 has a higher test accuracy than A2, that A1 will also have a higher score than A2.
>
> - *"How well does this generalize to larger models and more complex datasets?"*
>
> To show that NEAR can be successfully applied to different tasks, we added results from the TransNAS-Bench-101-Micro benchmark (Appendix Table 9) (see also our response to reviewer 6BBR). Due to time constraints, we were not able to run the computations with different proxies, but NEAR shows good performance compared to values reported in the literature. Additionally, we evaluated NEAR on HW-GPT-Bench, where NEAR outperforms MeCo$_\text{opt}$ and shows a correlation as high as the number of parameters. The results have been included in the appendix in Table 10.
>
> - *"For convolution operations, there are many hyperparameters like kernel size, stride, dilation, in-channels, out-channels, grouped convolutions, etc. How does the flattening of these matrices and calculating the score take into account all of the above factors? The intuition in this case is unclear."*
>
> The reviewer is right that different parameters for the convolution affect the performance. Because of the way the reshaping is done, the number of output channels goes directly into the score, since the matrix will have the same size as the number of channels. The number of input channels is identical to the number of output channels of the previous layer, so it is also included in the calculation. The influence of kernel size and stride is less direct. However, a small kernel size means that local information is used, i.e., the matrix has similar entries and the effective rank is low for inputs with few local differences. Most importantly, the rank calculated by sampling should show the same trends as the full matrix when the relevant parameters (such as the number of channels) are varied. We have added Figures A.6 and A.7 in the Appendix to show that the sampling strategy is able to capture the important behavior.
>
> - *"In the case of using ranks of pre and post-activation matrices as indicators of accuracy, a study that verifies this claim would be helpful. The same goes for the convolution part."*
>
> We are not aware of any study that analyze the pre- and post-activation matrices as indicators of good accuracy. However, our own intensive testing on the reported NAS benchmarks appears to support this claim. From a theoretical perspective, there is the connection to the number of linear regions (please see also our response to reviewer qDTu) that can be counted using the post-activation matrices. This gives at least some intuition why the post-activation matrices can be indicators of accuracy.
>
> - *"Specifically for the convolution part how accurate is the predicted rank using sampling? What is the minimum sample size required?"*
>
> We have observed that the rank calculated from the sampled matrix deviates from the rank obtained from the full matrix, as can be seen in the added Figures A.8 and A.9 in the Appendix. However, it is important to highlight that the absolute value of the rank is not the primary focus for our performance predictions. Instead, our interest lies in capturing the relative changes in rank as the number of channels is modified or as the kernel size changes. Our sampling scheme effectively captures these trends and variations, as demonstrated by the results presented in the aforementioned figures. Even with a single sample, the Spearman correlation between the full rank and the sampled rank is greater than 0.9. However, using multiple samples increases the correlation. More details can be found in the added Table 13 in the Appendix.
>
> - *"For the results, in cases where the estimations are off, why does that happen? A post-analysis would provide some insights into cases where NEAR is not optimal"*
>
> We did not find a clear correlation between model architecture and poor NEAR performance. However, similar to most other proxies, the prediction seems to be most often poor for models with many parameters. Unlike other proxies such as ZiCo, NEAR's hundred worst predictions do not include networks for which a high accuracy was incorrectly predicted (see added Figure A.6 in the appendix).

---

> > ### Comment · Reviewer_o26N · 2024-11-27
> >
> > Thank you for the detailed response. It satisfies all my remarks.

---

> ### Author Response · Authors · 2024-11-26
>
> - *"It has been shown in the literature that randomly generated matrices in high dimensions contain almost linearly independent vectors, how does this impact the prediction if we only derive the rank hypothesis only from a few input samples?"*
>
> It is true that random matrices in high-dimensional space contain almost linearly independent vectors. However, it is important to note that linear independence is not a sufficient condition for a large effective rank. For example, the matrix
> $$
> \begin{bmatrix}
> 10^6 & 0 & 0 \\\\
> 0 & 1 & 0 \\\\
> 0 & 0 & 1
> \end{bmatrix}
> $$
> has a rank of $3$ but an effective rank close to $1$. Furthermore, we argue that the output of an untrained CNN is different from a random matrix. For example, untrained CNNs have been used as feature extractors in previous works [1].
>
> [1] Z. Tong, G. Tanaka, "Reservoir Computing with Untrained Convolutional Neural Networks for Image Recognition," 24th International Conference on Pattern Recognition (ICPR) 2018.

---

### Official Review · Reviewer_qDTu · 2024-11-04

**Soundness:** 2
**Presentation:** 2
**Contribution:** 2
**Rating:** 5
**Confidence:** 4

**Summary:**

This paper introduces NEAR (Network Expressivity by Activation Rank), a zero-cost proxy (ZCP) that estimates the rank of neural network architectures via the effective rank of pre- and post-activation matrices at initialization (without training). The method is quite generic and is agnostic to the type of activation function or labelled data. Experiments on benchmarks such NAS-Bench-101 and NATS-Bench demonstrate NEAR’s high correlation with model accuracy, often outperforming other proxies.

**Strengths:**

I think the direction of using the effective matrix rank as a zero-cost proxy (ZCP) for neural net accuracy is a useful idea and worth pursuing. Some of the results are interesting and the NEAR proxy seems to filter out architectures that perform worst and does provide somehow reliable information on which architectures are in the top 10% best performing ones.

**Weaknesses:**

Despite the aforementioned positives, I have the following major concerns related to this submission:

- The differences in test loss in Table 7 are not statistically significant and it seems to me that even for these small networks the NEAR score can't determine correctly the ranking of the models. Furthermore, there are only a few tasks and datasets considered for this evaluation, and despite NEAR being able to filter out the bad architectures, typically one is interested in finding the best architecture among the top 10% of architectures in the search space, which this is clearly not the case from the results in Table 8.

- I think that the usage of the standard image classification benchmarks in NAS such as NATS-Bench or NAS-Bench-101, as in this paper, poses the risk of overfitting new proposed methods on these benchmarks. For instance, I would recommend the authors running their method on benchmarks of NAS-Bench-Suite-Zero [1].

- The contributions of the proposed approach are marginal regarding novelty, and despite I find the idea behind using effective rank useful, I still do not fully understand how it can be used to determine effective training and generalization of neural nets at initialization. Can the authors comment on this point further (see questions below as well)?

- In terms of significance, I do believe that the zero-cost proxy research could be beneficial to the community if it is rephrased in the current landscape of LLM compression [2]. I do find the standard NAS benchmarks useful for prototyping accuracy proxies, however I do not see anymore the practical relevance of these benchmarks outside prototyping and fast experimentation. This is my personal opinion though and my recommendation would be to think of ways how ZCP research on CNN models could be rephrased to structural pruning of LLMs (actually the ZCPs were inspired from network pruning methods in the first place).

- Finally, I think the paper could be made easier to read and gain more clarity with less text and a few more figures summarizing some of the main results described in the text.

-- References --

[1] Krishnakumar et al. NAS-Bench-Suite-Zero: Accelerating Research on Zero Cost Proxies. In NeurIPS 2022 DBT

[2] Zhu et al. A Survey on Model Compression for Large Language Models. In TACL 2024

**Questions:**

- What is the cost of computing NEAR and how does it compare to the cost of computing the other ZCPs?

- Can the authors comment on the relationship between their findings on the optimal initializaiton scheme and activation functions, and the maximal update parameterization (muP) [1], which enables effective transfer of hyperaparameters across model scales?

- As the efficiency of transformer-based models is becoming more and more relevant as their capacity keeps growing, I would suggest the authors assess the performance of their proxy method on approximating the rank of pruned transformer models. Could the authors evaluate their method on the recent HW-GPT-Bench [2]?

- Do the authors have any more theoretical justification of the effective matrix rank and its ability to assess the neural network generalization, trainability and expressivity at initialization? Is the effective rank of the pre- and post-activation matrices somehow related to the number of linear separable regions in TE-NAS [3]?

-- References --

[1] Yang et al. Tensor Programs V: Tuning Large Neural Networks via Zero-Shot Hyperparameter Transfer. 2022

[2] Sukthanker et al. HW-GPT-Bench: Hardware-Aware Architecture Benchmark for Language Models. In NeurIPS 2024 DBT

[3] Chen et al. Neural Architecture Search on ImageNet in Four GPU Hours: A Theoretically Inspired Perspective. In ICLR 2021

---

> ### Author Response · Authors · 2024-11-26
>
> - *"The differences in test loss in Table 7 are not statistically significant and it seems to me that even for these small networks the NEAR score can't determine correctly the ranking of the models. Furthermore, there are only a few tasks and datasets considered for this evaluation, and despite NEAR being able to filter out the bad architectures, typically one is interested in finding the best architecture among the top 10 of architectures in the search space, which this is clearly not the case from the results in Table 8."*
>
> We agree that the differences in test loss in Table 7 are small. However, it is important to note that the focus of this comparison is on a fixed architecture, where only the weight initialization scheme and activation function are varied. Given these subtle changes, the small differences in loss are expected. While NEAR may not always predict the absolute best combination, its strength lies in its ability to consistently filter out badly performing settings. While many tasks aim at finding the global best setting, there are also many tasks that require to find several good settings, for example to generate an ensemble model. To further demonstrate the effectiveness of NEAR in identifying good weight initialization schemes and activation functions, we have included Table 11 in the Appendix that compares the predictions of NEAR with those of other zero-cost proxies. This comparison shows that NEAR performs better in this specific task.
>
> - *"I think that the usage of the standard image classification benchmarks in NAS such as NATS-Bench or NAS-Bench-101, as in this paper, poses the risk of overfitting new proposed methods on these benchmarks. For instance, I would recommend the authors running their method on benchmarks of NAS-Bench-Suite-Zero [1]."*
>
> We agree with your concerns about the potential risk of overfitting newly proposed methods on these benchmarks, and we appreciate your suggestion to evaluate our approach on a wider variety of tasks. In response to your recommendation, we have extended our experiments to include 7 tasks from the TransNAS-Bench-101-Micro benchmark. Due to time constraints, we have only calculated the correlations for our proxy. However, when we compare to values reported in the literature, NEAR shows promising performance on these additional tasks. The results have been included in the Appendix in Table 9.
>
> - *"The contributions of the proposed approach are marginal regarding novelty, and despite I find the idea behind using effective rank useful, I still do not fully understand how it can be used to determine effective training and generalization of neural nets at initialization. Can the authors comment on this point further (see questions below as well)?"*
>
> We appreciate the reviewer's comment and would like to clarify the novelty and utility of our approach. While the concept of the effective rank has been explored in previous work, our contribution lies in its application to analyze neural networks at initialization, which represents a novel extension. Furthermore, we introduce a sampling scheme that enables the effective rank to be applied to convolutional layers, as described in Subsection 3.3.
>
> The effective rank is a metric that captures several key aspects that we hypothesize are connected to well-performing networks. Specifically, it measures the differences in activation patterns, where larger differences indicate better network performance. Additionally, it penalizes activation patterns with large differences in magnitude, which can lead to vanishing or exploding gradients. Moreover, connections can be drawn between the effective rank and the number of linear regions, as discussed in our response below.

---

> ### Author Response · Authors · 2024-11-26
>
> - *"In terms of significance, I do believe that the zero-cost proxy research could be beneficial to the community if it is rephrased in the current landscape of LLM compression [2]. I do find the standard NAS benchmarks useful for prototyping accuracy proxies, however I do not see anymore the practical relevance of these benchmarks outside prototyping and fast experimentation. This is my personal opinion though and my recommendation would be to think of ways how ZCP research on CNN models could be rephrased to structural pruning of LLMs (actually the ZCPs were inspired from network pruning methods in the first place)."*
>
> While we agree that extending our research to explore LLM compression is indeed a promising direction, we believe that there is still significant value in developing new performance predictors. In particular, our approach can be applied beyond pure NAS settings. For example, we have shown that NEAR is effective in identifying good activation functions and weight initialization schemes. These are critical tasks for various machine learning models and they are not limited to NAS applications. In addition, we have extended our evaluation to include benchmarks on LLMs, as detailed in Table 10 of the Appendix. These results demonstrate that NEAR is also applicable and beneficial in the context of LLMs, thereby broadening the scope and relevance of our research.
>
> - *"Finally, I think the paper could be made easier to read and gain more clarity with less text and a few more figures summarizing some of the main results described in the text."*
>
> We agree that figures would be a good addition and make our paper easier to follow. However, due to the strict length limit, it is not possible to include as many figures as one would like. Nevertheless, we have taken the reviewer's comment into account and added a few more figures and tables to the appendix to provide additional results that support our findings without much text.
>
> - *"What is the cost of computing NEAR and how does it compare to the cost of computing the other ZCPs?"*
>
> To compute the NEAR score, we only need to compute one forward pass through the network. Thanks to the reshaping operation for CNNs, a single-sample forward pass is sufficient for most convolutional layers. For fully connected layers, we need a forward pass with as many samples as there are neurons. Then, we need to compute the singular values for each layer, which has a complexity of $\mathcal{O}(n^3)$, where $n$ is the number of neurons or the number of channels for convolutional layers. Similarly, MeCo computes the eigenvalues of an $n \times n$ matrix, where $n$ is also the number of neurons or channels, depending on the layer type. The computation of eigenvalues has the same complexity of $\mathcal{O}(n^3)$, making MeCo's complexity essentially equivalent to that of NEAR. However, MeCo_opt uses only a subset of 8 channels and thus reduces the complexity.
>
> - *"Can the authors comment on the relationship between their findings on the optimal initializaiton scheme and activation functions, and the maximal update parameterization (muP) [1], which enables effective transfer of hyperaparameters across model scales?"*
>
> We see that the referenced work could be interesting for comparison and further studies. However, at the current stage, we cannot draw any connections to the findings of our work.
>
> - *"As the efficiency of transformer-based models is becoming more and more relevant as their capacity keeps growing, I would suggest the authors assess the performance of their proxy method on approximating the rank of pruned transformer models. Could the authors evaluate their method on the recent HW-GPT-Bench [2]?"*
>
> Thanks for the excellent suggestion to try NEAR on the HW-GPT Bench. We randomly sampled 20 000 architectures from the small variant of the search space and computed the correlation between NEAR and the perplexity scores. For comparison, we also computed the correlation of MeCo_opt. NEAR outperforms MeCo_opt and shows roughly the same correlation as the number of parameters. The results have been included in the Appendix in Table 10.

---

> ### Author Response · Authors · 2024-11-26
>
> - *"Do the authors have any more theoretical justification of the effective matrix rank and its ability to assess the neural network generalization, trainability and expressivity at initialization? Is the effective rank of the pre- and post-activation matrices somehow related to the number of linear separable regions in TE-NAS [3]?"*
>
> The number of binary activation patterns (i.e., 1 means the neuron is on, 0 means the neuron is off) is one way to estimate the number of linear regions of ReLU networks. While the number of linear regions is only lower if two activation patterns are exactly the same, the effective rank will already lower the value of the post-activation matrix if the patterns are similar. In this way, NEAR captures some notion of trainability. For example, let us consider two ReLU units with two inputs and all weights equal to $0.5$. If we allow only permit non-negative inputs, this network has two linear regions. One containing only $(0, 0)$ and one containing the remaining of $\mathbb{R}^2_{>0}$. Now, multiplying some weights by a large number does not change the number of linear regions, but it lowers the effective rank of the post-activation matrix. Since it is more difficult to train the network with weights of very different sizes, the effective rank can capture some notion of trainability.

---

### Official Review · Reviewer_6bBR · 2024-11-04

**Soundness:** 3
**Presentation:** 3
**Contribution:** 1
**Rating:** 5
**Confidence:** 4

**Summary:**

Zero-cost proxies in Neural Architecture Search aim to predict the performance of neural networks on a target task without training them (hence, zero-cost). The authors point out that the current zero-cost proxies share a few common limitations. First, they show a weaker correlation with the network performance than the number of parameters, which is a naive baseline. Second, they require a pre-defined search space, which may be unclear in many machine learning applications. Third, they have only been applied for finding optimal architectures, and they cannot be used to tune or select other important hyperparameters, such as the activation function and weight initialization scheme. To overcome these limitations, the authors propose a new zero-cost proxy called NEAR (Network Expressivity by Activation Rank) that estimate the optimal layer size for MLP networks without a search space. It can also be used to tune other miscellaneous hyperparameters. The idea behind NEAR consists of two key technical contributions 1) relaxing the definition of the activation patterns to encompass non-ReLU activation functions and 2) utilizing the effective rank of pre- and post-activation matrices to compute the NEAR score. The effectiveness of NEAR is verified empirically through experiments on several NAS benchmarks.

**Strengths:**

- The authors propose a novel zero-cost proxy score that can be utilized to predict not only the optimal architecture of the neural network but also the optimal set of hyperparameters, such as the weight initialization scheme and activation functions.

- The proposed network utilizes the effective rank of pre- and post-activation matrices to predict the final performance of a neural network, which appears to be theoretically sound. Such an approach, however, has been used in previous literature, as noted by the authors in lines 152-154. Therefore, the main technical contribution of NEAR is relaxing the definition of the activation patterns to include various activation functions.

- The authors present sufficient empirical evidence to show the effectiveness of the proposed approach from three main perspectives: 1) high rank correlation on NAS benchmarks, 2) optimal layer size estimation, and 3) activation function and weight initialization selection.

**Weaknesses:**

- Since the proposed method can be used to predict the performance of non-ReLU-based architectures, it can theoretically be applied to predict the performance of LLMs with GELU activations. Have the authors attempted this? It would be a strong extension of the proposed method especially since the authors argue that the major advantage of the proposed method is its applicability to tasks that do not have a fixed search space.

- The authors argue that this method enables estimation of optimal weight initialization and activation function, but it was not clear after reading the paper why other zero-cost proxies cannot be used for this purpose. It seems that although previous metrics did not explicitly take weight initialization or activation function into consideration, they can theoretically be used for the same purpose. Therefore, it would be helpful if the authors could add a comparison with previous approaches in Tables 7 and 8 to better highlight that only NEAR score can do this.

- I didn't quite follow how relaxing the definition of activation patterns affects the definition of the NEAR score. If the definition isn't relaxed and is constrained only to ReLU activations, how would the NEAR score be formulated? Or would it be impossible to formulate this score in the first place? If there is no difference between the two (theoretically speaking), why did this definition have to be relaxed? Does the NEAR score before and after relaxing this definition show lower/higher correlation with performance? Because this is unclear, I didn't understand how the NEAR is designed to serve all of its intended purposes, such as choosing the optimal number of layers, weight initialization, and activation function.

- In Tables 1-3, The NEAR score only outperforms other metrics only on NAS-Bench-101 and NATS-Bench-SSS (CIFAR-100). Tables 5-8 do not compare the NEAR score against previous approaches. This makes me question whether the NEAR score is as reliable as the authors claim. I would like to see at least one application in which the NEAR score is clearly far more advantageous than other metrics.

In general, the authors make very strong arguments about the theoretical novelty of the proposed proxy and its usability. However, because of the points stated above, I did not find the current manuscript convincing enough. After reading this paper, I am left wondering in what exact practical application the NEAR score would actually be more useful than previous proxies. If the authors can explicate both theoretically and empirically that the NEAR score does indeed have distinct advantages, I would be willing to adjust my score.

**Questions:**

Please refer to the Weaknesses section.

---

> ### Author Response · Authors · 2024-11-26
>
> - *"Since the proposed method can be used to predict the performance of non-ReLU-based architectures, it can theoretically be applied to predict the performance of LLMs with GELU activations. Have the authors attempted this? It would be a strong extension of the proposed method especially since the authors argue that the major advantage of the proposed method is its applicability to tasks that do not have a fixed search space."*
>
> While NEAR can indeed be applied to predict the performance of networks using the GELU activation, our statement about not needing a search space was specifically for multi-layer perceptrons (line 66). Still, we agree that applying NEAR to large language models (LLMs) with GELU activations would be a worthwhile extension of our method. Based on this suggestion, we sampled 20 000 random architectures from the small version of the HW-GPT-Bench benchmark and calculated the correlation with the perplexity. Due to time constraints, we only compare NEAR to MeCo_opt. NEAR appears to be as good as the number of parameters and outperforms MeCo_opt. These results have been included in the Appendix in Table 10.
>
> - *"The authors argue that this method enables estimation of optimal weight initialization and activation function, but it was not clear after reading the paper why other zero-cost proxies cannot be used for this purpose. It seems that although previous metrics did not explicitly take weight initialization or activation function into consideration, they can theoretically be used for the same purpose. Therefore, it would be helpful if the authors could add a comparison with previous approaches in Tables 7 and 8 to better highlight that only NEAR score can do this."*
>
> The reviewer is correct that, in theory, other zero-cost proxies could be adapted for estimating optimal weight initialization and activation function, even if they were not explicitly designed for these tasks. However, as we also note in the manuscript, many existing proxies are inherently limited to specific activation functions, such as ReLU. By repeating the experiments reported in Table 8 for all proxies, we show that NEAR performs better than all other proxies tested in detecting good weight initialization schemes and activation functions. Note that reg\_swap and swap are not included in the analysis because they are limited to the ReLU activation function. We added a table including this information in the Appendix (Table 11).

---

> ### Author Response · Authors · 2024-11-26
>
> - *"I didn't quite follow how relaxing the definition of activation patterns affects the definition of the NEAR score. If the definition isn't relaxed and is constrained only to ReLU activations, how would the NEAR score be formulated? Or would it be impossible to formulate this score in the first place? If there is no difference between the two (theoretically speaking), why did this definition have to be relaxed? Does the NEAR score before and after relaxing this definition show lower/higher correlation with performance? Because this is unclear, I didn't understand how the NEAR is designed to serve all of its intended purposes, such as choosing the optimal number of layers, weight initialization, and activation function."*
>
> If the definition of the activation pattern were not relaxed, the post-activation matrix would be a binary matrix, where a $1$ refers to an active ReLU unit and a $0$ to an inactive one. However, this definition would limit the applicability of the NEAR score to ReLU activation functions only, since it is unclear how to define an "active" unit for other activation functions, such as Tanh or Sigmoid. Furthermore, we could only use the post-activation matrix, as again, active and inactive are not well defined concepts without ReLU. By considering the magnitude of the pre-activation and post-activation values, we can capture the degree of activation for different types of activation functions, including Tanh and Sigmoid. Because we can capture smaller differences in activation values, this also allows us to compare different activation functions and weight initialization schemes. As more and more neurons are added to a layer, the network will be unable to fully utilize some of those neurons, and the magnitude of their activations will be similar across samples. This allows NEAR to estimate a good number of neurons for a layer (see Section 4.2). Below is a table comparing the correlation achieved with and without relaxing the definition. As you can see, relaxing the definition improves the correlation on all benchmarks and datasets tested, with the exception of ImageNet16-120 on NATS-Bench-TSS, where the correlation remained the same.
>
> | Search space         | NATS-Bench-TSS |      |           |      |          | NATS-Bench-SSS |          |      |           |      |
> | -------------------- | -------------- | ---- | --------- | ---- | -------- | -------------- | -------- | ---- | --------- | ---- |
> | Dataset              | CIFAR-10       |      | CIFAR-100 |      | IN16-120 |                | CIFAR-10 |      | CIFAR-100 |      | IN16-120 |  |
> | Correlation | τ              | ρ    | τ         | ρ    | τ        | ρ              | τ        | ρ    | τ         | ρ    | τ | ρ |
> | Not relaxed          | 0.66           | 0.85 | 0.67      | 0.86 | 0.66     | 0.84           | 0.73     | 0.9  | 0.51      | 0.7  | 0.68 | 0.86 |
> | NEAR                 | 0.7            | 0.88 | 0.69      | 0.87 | 0.66     | 0.84           | 0.74     | 0.91 | 0.62      | 0.82 | 0.76 | 0.92 |
>
> - *"In Tables 1-3, The NEAR score only outperforms other metrics only on NAS-Bench-101 and NATS-Bench-SSS (CIFAR-100). Tables 5-8 do not compare the NEAR score against previous approaches. This makes me question whether the NEAR score is as reliable as the authors claim. I would like to see at least one application in which the NEAR score is clearly far more advantageous than other metrics."*
>
> Tables 5 and 6 refer to the observation that the layer-wise NEAR score shows a scaling behavior that allows the estimation of a good layer size. No such scaling could be observed for MeCo$_\text{opt}$. Other proxies were not investigated. For the determination of good weighting initialization schemes or activation functions (Table 8), we have now added a comparison with other proxies in Table 11 in the appendix. NEAR shows the highest correlation of all proxies tested.
>
> We have also added results from the TransNAS-Bench-101 Micro benchmark in Table 9 in the Appendix, where NEAR performed well across all tasks tested. Due to time constraints, we were not able to replicate experiments from the literature, but compared to previously published results, NEAR performs well.

---

### Official Review · Reviewer_he4q · 2024-11-04

**Soundness:** 3
**Presentation:** 3
**Contribution:** 2
**Rating:** 6
**Confidence:** 4

**Summary:**

This paper presents a method for identifying optimal architectures in Neural Architecture Search (NAS) using a zero-cost proxy score derived from the activation rank, without requiring separate training. To achieve this, the authors calculate both pre- and post-activation ranks and introduce a sampling method to approximate matrices for efficiency. Additionally, the paper thoroughly discusses the potential of the proposed approach using effective rank.

**Strengths:**

* **Applicability Across Architectures**: A key contribution of this work is that the proxy score is applicable regardless of architecture structure or activation function, providing robustness and versatility.
* **Benchmark Performance**: The authors demonstrated the method’s efficacy across standard NAS benchmarks, showing its comparative strength against other zero-cost proxies.
* **Efficiency**: By not requiring separate training, the method significantly reduces computational overhead, which is valuable for practical NAS applications.

**Weaknesses:**

* **Lack of Theoretical Clarification**: While the experimental results demonstrate the effectiveness of the NEAR score, the theoretical significance of using the effective rank (erank) in Definition 3.2 remains somewhat unclear. Numerous NAS studies have previously explored concepts related to rank. It would strengthen the paper if the authors could provide a more detailed theoretical analysis comparing NEAR to existing methods, highlighting any novel insights their approach offers. Additionally, clarifying why redefining Definition 3.1 is necessary or beneficial in this context would enhance the theoretical foundation of the work.

* **Evaluation Across Layers**: It is not entirely clear whether calculating the NEAR score across all layers is necessary to achieve optimal performance. The paper does not present experiments exploring how the NEAR score changes when calculated on different subsets of layers. Conducting such experiments could provide insights into the contributions of individual layers to the overall score and discuss the implications for computational efficiency and accuracy.

* **Comparison with Few-Shot Methods**: The paper focuses on zero-shot performance but does not compare the NEAR score with few-shot methods that involve minimal training (e.g., a few epochs). As noted in "Demystifying the Neural Tangent Kernel from a Practical Perspective: Can it be trusted for Neural Architecture Search without training?" (Mok et al., 2022), it would be beneficial to conduct experiments comparing the zero-shot NEAR score to scores obtained after a few epochs of training. This comparison could substantiate the robustness of the NEAR score and reinforce the claim that it can effectively identify optimal neural networks without training.

**Questions:**

* **Theoretical Advantages over Existing Methods**: Could the authors provide a more detailed theoretical analysis comparing the NEAR score to existing zero-cost proxies? Specifically, what theoretical advantages does the NEAR score offer that may account for its superior performance? Highlighting any novel insights or mechanisms by which NEAR improves upon prior methods would strengthen the contribution.

* **Justification for Redefining Definition 3.1**: Since Definition 3.1 (Effective Rank) was established in previous studies, can the authors clarify why redefining it here is necessary or beneficial in the context of their method? Providing justification or modifications that are specific to their approach would help readers understand the significance of this definition in their work.

* **Layer-wise Contribution to NEAR Score**: To understand the impact of each layer on the NEAR score, could the authors conduct experiments exploring how the score changes when calculated on different subsets of layers? For instance, does excluding certain layers significantly affect the correlation with final model accuracy? Discussing these findings could provide insights into optimizing computational efficiency without compromising accuracy.

* **Stability of NEAR Score After Minimal Training**: In line with observations from Mok et al. (2022) (1), have the authors considered evaluating the stability of the NEAR score after a few epochs of training? Conducting experiments that compare the zero-shot NEAR score to scores obtained after 2-3 epochs could demonstrate the robustness of the method and whether minimal training affects its predictive capabilities.

* **Comparison with Few-Shot Performance**: How does the zero-shot NEAR score compare with few-shot methods in terms of computational cost and accuracy? Including a comparison could help position the NEAR score within the broader context of NAS methods and provide evidence for its practical advantages.

**References**
1. “Demystifying the Neural Tangent Kernel from a Practical Perspective: Can it be trusted for Neural Architecture Search without training?” (Mok et al., 2022)

---

> ### Author Response · Authors · 2024-11-26
>
> - *"Theoretical Advantages over Existing Methods: Could the authors provide a more detailed theoretical analysis comparing the NEAR score to existing zero-cost proxies? Specifically, what theoretical advantages does the NEAR score offer that may account for its superior performance? Highlighting any novel insights or mechanisms by which NEAR improves upon prior methods would strengthen the contribution."*
>
> Section 3.2 qualitatively explains why the effective rank of the pre- and post-activation matrix contains information about the accuracy of the network. NEAR employs the effective rank of both pre- and post-activation matrices to capture the diversity of activations in response to a given input. We argue that a higher effective rank implies a richer representation space and indicates that the network is using its capacity more effectively. We hypothesize that the effective rank is a proxy for the information capacity of the network at a given layer and across the network. Intuitively, a network with a higher effective rank is capable of representing more complex functions. We have shown experimentally on several NAS benchmarks that NEAR really correlates with accuracy after training. One can also draw a connection to the number of linear regions, as outlined in our response to reviewer qDTu. However, a detailed mathematical analysis is beyond the scope of this work and open for future work.
>
> - *"Justification for Redefining Definition 3.1: Since Definition 3.1 (Effective Rank) was established in previous studies, can the authors clarify why redefining it here is necessary or beneficial in the context of their method? Providing justification or modifications that are specific to their approach would help readers understand the significance of this definition in their work."*
>
> We appreciate you raising the point about the restatement of Definition 3.1 (Effective Rank). While the definition remains consistent with previous work, we think it is important to restate it here to improve readability and self-containedness.
>
> - *"Layer-wise Contribution to NEAR Score: To understand the impact of each layer on the NEAR score, could the authors conduct experiments exploring how the score changes when calculated on different subsets of layers? For instance, does excluding certain layers significantly affect the correlation with final model accuracy? Discussing these findings could provide insights into optimizing computational efficiency without compromising accuracy."*
>
> We conducted experiments to show the impact of each layer on the NEAR score. Specifically, we computed the Spearman correlation for over a billion subsets consisting of six layers each and aggregated these correlations for every layer within each subset. By averaging these correlation values, our goal was to determine which layers might play the most significant role in the predictions. However, our findings did not show substantial variations in the impact of individual layers. This indicates that the NEAR score likely relies not on specific layers alone, but on the combined contributions of all layers together. We added a figure of the average correlation in the Appendix (Figure A.7).
>
> - *"Stability of NEAR Score After Minimal Training: In line with observations from Mok et al. (2022) (1), have the authors considered evaluating the stability of the NEAR score after a few epochs of training? Conducting experiments that compare the zero-shot NEAR score to scores obtained after 2-3 epochs could demonstrate the robustness of the method and whether minimal training affects its predictive capabilities."*
>
> In response to your suggestion, we conducted experiments on the NATS-Bench-SSS benchmark using the CIFAR-10 dataset. We trained all networks for 1, 3, 5, and 10 epochs and evaluated the performance of various proxies, including NEAR. Our results show that NEAR shows a notable increase in correlation with the final network performance after only a few epochs of training. Interestingly, NEAR appears to be the only one of all the proxies tested that shows a meaningful improvement in its predictive ability within these early stages of training. We also note that the decrease in correlation after a single epoch seems to be consistent with the empirical observation that the number of linear regions decreases during early training before increasing again [1]. Unfortunately, due to time constraints, we were unable to extend these experiments to other search spaces and datasets as outlined in the original submission. We added a figure showing the changes in the correlation after 1, 3, 5, and 10 epochs in the Appendix (Figure A.5).
>
> [1] B. Hanin, D. Rolnick, "Complexity of Linear Regions in Deep Networks", 36th International Conference on Machine Learning (ICML) 2019.

---

> ### Author Response · Authors · 2024-11-26
>
> - *"Comparison with Few-Shot Performance: How does the zero-shot NEAR score compare with few-shot methods in terms of computational cost and accuracy? Including a comparison could help position the NEAR score within the broader context of NAS methods and provide evidence for its practical advantages."*
>
> Unfortunately, due to the time constraints of this review process, we were unable to perform a comprehensive comparison between few-shot methods and our zero-cost proxy. However, we can provide some insights into the computational cost of calculating NEAR. This cost is significantly lower than training of one or several supernets. To calculate the NEAR score, we only need to compute one forward passes through the network. Thanks to the reshaping operation for CNNs, a forward pass with a single sample is sufficient for most convolutional layers. For fully connected layers, we need a forward pass with as many samples as there are neurons. Subsequently, we need to compute the singular values for each layer, which has a complexity of $\mathcal{O}(n^3)$, where $n$ is the number of neurons or the number of channels for convolutional layers.

---

> > ### Comment · Reviewer_he4q · 2024-11-28
> >
> > I sincerely appreciate the authors' detailed explanations and additional experiments in response to the raised questions. However, some concerns remain unresolved.
> >
> > ### **Theoretical Foundation of NEAR Score (Section 3.2)**:
> > The theoretical justification provided in response to the query regarding Section 3.2 still does not adequately address the core issue. For NEAR to serve as a proxy for the information capacity of the network, the definition and its implications require further experimental validation and theoretical support. Merely demonstrating a correlation between the effective rank and the final accuracy after training does not sufficiently substantiate the claim. The logical leap from observed correlation to theoretical significance seems overly optimistic. Without a stronger theoretical grounding, the paper’s potential impact on the community may appear more limited than expected. In particular, if the definition presented does not yield more general insights into neural networks, the contribution of this paper to the community is significantly constrained.
> >
> > ### **Layer-Wise Contribution to NEAR Score**:
> > The claim that all layers contribute equally to the NEAR score also raises concerns. If the NEAR score is proposed as a proxy for network capacity, it is reasonable to expect that individual layers would contribute less to the overall score than the combined score across the entire network. If the overall NEAR score indeed serves as a stronger proxy, the correlation calculated using the full network should demonstrably outperform those calculated using individual layers. Without experimental results to support this, the explanation of the NEAR score remains incomplete and difficult to reconcile with its theoretical justification.
> >
> > Given the unresolved disconnect between the theoretical justification and the experimental results, I find it difficult to fully accept the claims regarding NEAR's contribution. Despite the authors' extensive efforts, I believe that the theoretical limitations and the inconsistencies in the experimental results constrain the broader impact of this work. Therefore, I will maintain my current review score.

---

> > > ### Author Response · Authors · 2024-12-03
> > >
> > > - *"Theoretical Foundation of NEAR Score (Section 3.2): The theoretical justification provided in response to the query regarding Section 3.2 still does not adequately address the core issue. For NEAR to serve as a proxy for the information capacity of the network, the definition and its implications require further experimental validation and theoretical support. Merely demonstrating a correlation between the effective rank and the final accuracy after training does not sufficiently substantiate the claim. The logical leap from observed correlation to theoretical significance seems overly optimistic. Without a stronger theoretical grounding, the paper’s potential impact on the community may appear more limited than expected. In particular, if the definition presented does not yield more general insights into neural networks, the contribution of this paper to the community is significantly constrained."*
> > >
> > > We agree that additional theoretical grounding would further strengthen our work. In response, we would like to highlight a connection to the phenomenon of rank collapse that has been observed in the literature, particularly in the context of batch normalization. Previous studies have shown that the initial rank of network activations significantly influences the learning dynamics and correlates with final accuracy (see, for example, Figure 1 in Reference [1]). This observation aligns with our findings and provides indirect support for our use of NEAR.
> > >
> > > [1] H. Daneshmand, J. Kohler, F. Bach, T. Hofmann, A. Lucchi, "Batch Normalization Provably Avoids Rank Collapse for Randomly Initialised Deep Networks", Adv. Neural Inf. Process. Syst. 2020, 33, 18387-18398.
> > >
> > > - *"Layer-Wise Contribution to NEAR Score: The claim that all layers contribute equally to the NEAR score also raises concerns. If the NEAR score is proposed as a proxy for network capacity, it is reasonable to expect that individual layers would contribute less to the overall score than the combined score across the entire network. If the overall NEAR score indeed serves as a stronger proxy, the correlation calculated using the full network should demonstrably outperform those calculated using individual layers. Without experimental results to support this, the explanation of the NEAR score remains incomplete and difficult to reconcile with its theoretical justification."*
> > >
> > > We thank the reviewer for the feedback on our revision. However, we think that there is a misunderstanding regarding the layer-wise contributions. As shown in Figure A.7, the averaged Spearman's correlation of NEAR scores calculated from individual layers is around 0.7, while Table 2 shows that the Spearman's correlation is larger than 0.9 if the overall NEAR scores of the full networks are employed. Consequently, the expectation of the reviewer (and us) is confirmed that the correlation of the full network's NEAR score demonstrably outperforms those of NEAR scores calculated using only individual layers. It is important to note that the layers in the NATS-Bench-SSS networks differ (due to variations in the number of channels), which means that the assignment of layer-wise contributions is not entirely precise. Nevertheless, it allows a comparison with the values in Table 2. Consequently, we would like to ask the reviewer for a reevaluation of the rating.

---

### Meta-Review · Area_Chair_2QGq · 2024-12-19

**Metareview:**

The paper presents a zero-cost proxy technique (NEAR) for predicting neural network performances. The proposed method leverages the effective rank of pre- and post-activation matrices to estimate model accuracy. The authors demonstrate good correlations across benchmarks like NAS-Bench-101 and NATS-Bench-SSS/TSS. The authors provide extensive experiments showing NEAR’s computational efficiency and robustness.

The paper is a borderline case and reviewers are mixed in their assessments of the merits and weaknesses of the work.

In terms of weaknesses, the theoretical basis for NEAR’s correlation with performance is underdeveloped, with only superficial connections to rank collapse and linear region theory. Its generalizability to large-scale architectures, such as transformers in LLMs, needs more validation, though preliminary results on HW-GPT-Bench are promising. The intuition behind the method’s handling of convolutional layers and its reliance on correlations without deeper ranking consistency analyses also leave room for improvement.

However, on the positive side, the paper offers a compelling empirical contribution with practical relevance, justifying acceptance despite areas needing further exploration.

As a result, I recommend accepting the paper, but I urge the reviewers to consider the comments in the camera ready.

**Additional Comments On Reviewer Discussion:**

The reviewers and three of the four authors engaged positively during the rebuttal. One reviewer (with a score of 5) did not engage during the rebuttal. The authors addressed positively most of the comments, therefore the AC acknowledges the authors' rebuttal efforts.

---

### Decision · Program_Chairs · 2025-01-22

Accept (Poster)